# EMERGENT PROPERTIES WITH REPEATED EXAMPLES

## ABSTRACT

We study the performance of transformers as a function of the number of repetitions of training examples with algorithmically generated datasets. On three problems of mathematics: the greatest common divisor, modular multiplication, and matrix eigenvalues, we show that for a fixed number of training steps, models trained on smaller sets of repeated examples outperform models trained on larger sets of single-use examples. We also demonstrate that *two-set training* - repeated use of a small random subset of examples, along normal sampling on the rest of the training set - provides for faster learning and better performance. This highlights that the benefits of repetition can outweigh those of data diversity. These datasets and problems provide a controlled setting to shed light on the still poorly understood interplay between generalization and memorization in deep learning.

## 1 INTRODUCTION

When training neural networks, it has become customary to use the largest and most diverse datasets available, and to limit example reuse as much as possible. This tendency is manifest in large language models. GPT (Radford & Narasimhan, 2018) was trained for 100 epochs (each example was seen 100 times on average), BERT (Devlin et al., 2019) on 40 and GPT-2 (Radford et al., 2019) on 20. In recent models, most examples in the pre-training corpus are seen only once, a few specialized datasets are iterated 2 or 3 times, and fine-tuning examples are seen once or twice. Meanwhile, data budgets are on the increase: GPT-2 was trained on less than 10 billion tokens, GPT-3 (Brown et al., 2020) was pre-trained on 300 billion, Chinchilla (Hoffmann et al., 2022) and Llama (Touvron et al., 2023) on 1.4 trillion, Llama2 (Touvron & et al., 2023) on 2 trillion, and Llama3 (Dubey & et al., 2024) on 15.6 trillion. Whereas the use of large train sets is grounded in theory (Vapnik & Kotz, 2006), the practice of not repeating training examples is less motivated. It reflects the belief that, when availability permits fresh data is superior to repeated use of a corpus (Komatsuzaki, 2019; Raviv et al., 2022; Hernandez et al., 2022; Muennighoff et al., 2023). This belief is grounded in the idea that memorization of repeated examples hinders generalization (Zhang et al., 2017). From a human learner point of view, this is counter-intuitive. When faced with a situation we never experienced, we *recall* similar instances (Proust, 1919), and use them as anchors to navigate the unknown. If memorization benefits human learners (Ambridge et al., 2015), why should it hinder machines?

In this paper we challenge the view that the repetition of training examples is undesirable, and that for a given *training budget* (TB, the *total* number of training examples), one should maximize the *data budget* (DB, the number of *distinct* training examples). We explore the impact of repeated samples in three controlled settings using generated data: computing the greatest common divisor (GCD) of two integers (Charton, 2024), modular multiplication of two integers, and calculating the eigenvalues of symmetric real matrices (Charton, 2022). These settings allow for perfect control over the distribution of repeated examples, unlike *natural datasets* (e.g. text from the web) which may feature unintended duplication and redundancy. Our experiments uncover two striking phenomena:

1. **Repetition Helps:** For fixed training budgets (300M to 1B examples), models trained from small data budgets (25 to 50M examples) outperform models trained on large DB. This sometimes gives rise to "emergent" phenomena: properties *only learned* by models trained on small DB.
2. **Two-Set Training:** For fixed data budgets, learning speed and performance are significantly enhanced by *randomly selecting* a subset of training examples, and repeating them more often during training. The "two-set effect" is all the more surprising as the repeated examples are not curated, and only differ from the rest of the training data by their frequency of use.

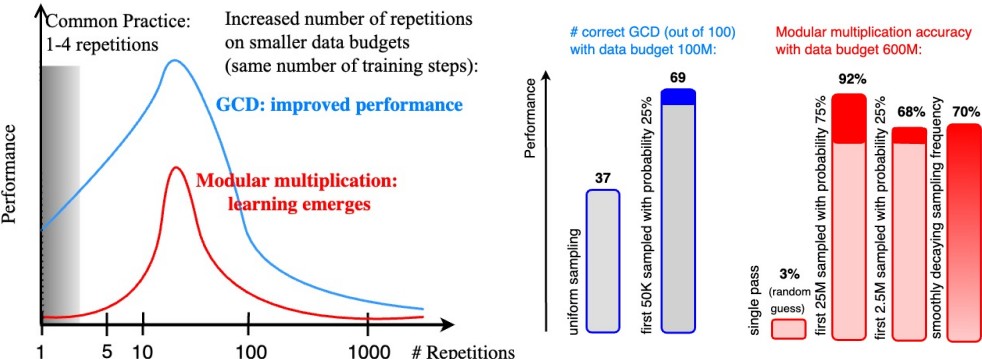

Figure 1: **Repetition Helps (Left):** Performance as a function of repetition for a fixed training budget (600M). *GCD (blue)*. Models trained on smaller datasets, repeated 30 times, perform much better than models trained on one to four epochs. *Multiplication mod 67 (red)*. Models trained for 1 to 4 epochs do not learn. Learning "emerges" when models are trained on smaller data budgets, with increased repetition; albeit in a setting where the number of examples is large enough to avoid overfitting.
**Two-set training (Right):** For a fixed data budget, splitting the data into two *random* subsets and increasing the training frequency of one greatly improves performance. *GCD (left)*: repeating 50k examples 3000 times for a training budget of 600M brings performance from 37 to 69 on 100M. *Modular multiplication (right)*: Models trained on 600M single-use examples do not learn. With 25M examples repeated 18 times, and 150M single use examples, accuracy is 92%, with 2.5M examples repeated 60 times, and 450M single-use, accuracy is 68%. Smooth distributions of repetition over the training set achieve 70% accuracy.

In ablation experiments, we show that the performance of two-set training cannot be improved by curating the set of repeated examples, or refreshing it as training proceeds. This sets us apart from *curriculum learning*, and strengthens the observation that repetition of a few *random examples* is really all we need. We also show that mixing repeated and non-repeated examples in the same mini-batches is required for two-set training to work. Finally, we propose a smooth extension of two-set training, by introducing a probability distribution on the training set.

Our work isolates an interesting phenomenon in a clean setting. The three tasks we study each exhibit idiosyncratic structure that allows to test a variety of hypotheses. For instance, the GCD dataset exhibits an inverse polynomial distribution of results, reminiscent of Zipf's law in natural language (Zipf, 1935). This allows us to test whether amplification of the tail of the distribution can benefit learning, by incorporating it into two-set training (while an attractive hypothesis, our ablations show that this seems not to be the case). In contrast, the results of modular multiplication are almost uniformly distributed, indicating that our conclusions do not depend on the existence of a power-law. Finally, the eigenvalue problem features non-linear, approximate calculations on reals.

In all three cases, the benefits of repetition are significant, but come in different flavors, from improving performance and accelerating learning (GCD), to allowing a new task to be learned (multiplication), or to be accessible to smaller models (eigenvalues). Alternatively, small random subsets of the data repeated at high frequency can elicit similar effects. These findings have profound implications and should lead to a paradigm shift where the training set size becomes a mere hyper-parameter, not solely governed by the availability of data and the belief that more is always better.

**Note.** *Training budget* is known as *compute budget* in other works (Power et al., 2022; Muennighoff et al., 2023). We use *training budget* to distinguish it from the compute cost arising from model size.

## 2 BACKGROUND AND RELATED WORK

In this paper, we focus on relatively small transformer models performing mathematical tasks, placing it into a long established corpus of works that study interesting phenomena in a controlled setting, and advance our understanding of the underlying mechanisms in larger models in the wild, see e.g. Power et al. (2022); Garg et al. (2022); Charton (2024); Dohmatob et al. (2024).

One such example is the study of *"grokking"*, first observed with modular arithmetic - a phenomenon where models generalize long after achieving 100% accuracy on their (small) training

set (Power et al., 2022; Liu et al., 2022b; 2023). On the surface, grokking shares similarities with our work: a small training dataset is iterated for many epochs, the phenomenon is isolated in clean experiments on synthetic data, and it contradicts traditional wisdom regarding overfitting (Mohri et al., 2018). But there are important differences: in grokking, delayed learning occurs, we observe no such delay; grokking occurs for "tiny" training samples (hundreds or thousands of examples), our models use millions (even for modular multiplication); grokking is very sensitive to the optimizer used, our findings are robust across optimizers (Appendix D.5), and, of course, no two-set approach is documented in the grokking setting.

Another related setting is *"benign overfitting"* (Bartlett et al., 2020; Belkin, 2021; Bartlett et al., 2021), where an *over-parametrized* model perfectly fits noisy data, without harming prediction accuracy. One could argue that our work presents a *quantitative* manifestation of benign overfitting, inasmuch as decreasing the data budget increases model over-parametrization. However, this would not account for the decrease in performance once the data budget falls below a certain number (one could argue that overfitting is no longer benign, then), nor for the possibility of two-set training.

Prior works have studied the role of data reuse in language models. Hernandez et al. (2022) study data repetition in models with up to 800M parameters with training budgets of 100M tokens to exhibit detrimental impact of repetition of a subset of the training data. In the context of scarcity of training data, Muennighoff et al. (2023) find for LLMs of contemporary size (up to 9B) that with constrained data for a fixed training budget, training with up to 4 epochs of repeated data yields negligible changes to loss compared to having unique data; any further repetition decreases the value of additional training. A limitation of these works is the lack of control over repetition in the training set: partial copies, of sentences, paragraphs sometimes whole documents, abound in pre-training corpora. Allen-Zhu & Li (2024) undertake a controlled study on synthetic language data in the context of knowledge *retrieval* and find that knowledge augmentation - repeated inclusion of reformulated variants - of a small subset of the data leads to performance improvement; an effect somewhat akin to what we observe in two-set training.

Our work is related to, but different from, *curriculum learning (CL)* (Bengio et al., 2009; Wang et al., 2022), where training data is presented in a meaningful order, usually from "easy" to "hard" samples. Two-set training differs from curriculum learning in at least two important ways: in CL, datasets are curated, our subsets are completely random; in CL, the training distribution shifts over time, while our subsets are static. Our ablations show that curating the repeated set, or changing it over time, as in CL, brings no improvement in performance (and may even have an adverse effect).

Lastly, our work touches upon the expansive area of *out-of-distribution (OOD)* generalization (Gulrajani & Lopez-Paz, 2021; Lopez-Paz, 2025), which studies generalization when train and test distributions differ. Curiously, while our two-set approach increases the frequency of some training examples, because the repeated set is chosen *at random*, the training set remains distributionally equivalent to the test set. Thus, our study falls outside the usual framework of OOD studies. For additional discussion of our setting and related work, see Appendix A.

## 3 EXPERIMENTAL SETTINGS AND BASELINES

We focus on three problems of mathematics: computing the greatest common divisor, multiplication modulo 67, and computing the eigenvalues of real symmetric matrices. The GCD and eigenvalues were studied in prior work (Charton, 2022; 2024; Dohmatob et al., 2024; Feng et al., 2024).

**Greatest common divisor.** The model is tasked to predict the GCD of two integers uniformly distributed between 1 and 1 million, encoded in base 1000. Following Charton (2024), who observes that throughout training almost all pairs of integers with the same GCD are predicted the same, we evaluate model performance by the number of GCD below 100 predicted correctly, measured on a random test sample of 100,000 pairs: 1000 pairs for each GCD from 1 to 100. Charton reports a best performance of 22 correct GCD for a model trained on uniformly distributed inputs.
**Note.** We prefer this test metric over a more standard accuracy on random input pairs, because the GCD are distributed according to an inverse square law. In particular the probability that a GCD is 1 is about 62%. As a result, the accuracy metric results in overly optimistic model performances.

**Modular multiplication.** Modular arithmetic plays an important role in many public key cryptography algorithms (Diffie & Hellman, 1976; Regev, 2005), and is known to be a hard problem for

neural networks (Palamas, 2017). Modular addition was studied in several previous works, in the context of grokking (Power et al., 2022; Liu et al., 2022a) and mechanistic interpretability (Zhong et al., 2023)[1]. While modular multiplication over $\mathbb{Z}/p\mathbb{Z}^{\times}$ is *mathematically* is equivalent to modular addition mod $p - 1$, these problems differ *computationally*, due to the hardness of the discrete logarithm (Diffie & Hellman, 1976). In most previous works on arithmetic modulo $p$, model inputs are sampled from integers between 0 and $p$, which results in a very small problem space for small $p$. In this work, we study the multiplication modulo 67 of two integers from 1 to 1 million. This allows for a much larger problem space, and training sets. Model accuracy is evaluated by the percentage of correct predictions of $a \times b \mod 67$, on a test set of $10,000$ examples (a new test set is generated at every evaluation). In this problem, all outcomes from 1 to 66 are uniformly distributed, while 0 appears nearly twice as often.

**Eigenvalue calculation.** This problem was introduced to deep learning by Charton (2022), who showed that transformers can learn to predict the eigenvalues of real symmetric matrices with independent and identically distributed entries, rounded to three significant digits. The eigenvalue problem is arguably a harder problem than the previous two, non-linear and typically solved by iterative algorithms. Nonetheless the eigenvalue problem seems an *easier* task for transformers, as even smaller transformers are able to solve this task for matrices of size up to 8x8 (Charton, 2022). We include this problem to show that our conclusions extend beyond arithmetic problems on integers. Note also that because matrix entries and eigenvalues are rounded, this problem features *noisy* inputs and outputs. Model accuracy is evaluated as the percentage of model predictions that predict the correct eigenvalues of a test matrix with less than 5% relative error (in $\ell^1$ distance). It is measured on a test set of $10,000$ samples, generated afresh at every evaluation.

**Models and tokenizers.** In all experiments, we use sequence-to-sequence transformers (Vaswani et al., 2017) with 4 layers in the encoder and decoder (4-layers encoders and 1-layer decoder for eigenvalues), an embedding dimension of 512, and 8 attention heads. Models have 35 million parameters for GCD and modular multiplication, and 22 million for eigenvalues. They are trained to minimize a cross-entropy loss, using the Adam optimizer (Kingma & Ba, 2014), with a learning rate of $10^{-5}$, over batches of 64. The integer inputs and outputs of the GCD and multiplication problems are tokenized as sequences of digits in base 1000, preceded by a separator token. The real numbers in the eigenvalue problem are encoded as floating point numbers, rounded to three significant digits, and tokenized as the triplet $(s, m, e)$ – sign, (base 1000) mantissa, (base 10) exponent – i.e. $f = s \cdot m \cdot 10^e$ (P1000 encoding from Charton (2022)). All experiments are run on one NVIDIA V100 GPU with 32 GB of memory.

## 4 REPETITION HELPS

We now embark on a systematic study of the impact of data budget on performance, for various training budgets. In other words, we compare the performance of models trained on datasets with a fixed number of examples (data budget), for increasing amounts of time (training budget).

On the **GCD problem,** we consider data budgets of $1, 5, 10, 25, 50$ and 100M distinct examples, and an "unlimited data" setting, where new examples are generated on the fly and DB$\approx$ TB[2]. For each data budget, we train 5 models with a training budget of over 1 billion examples, and report their average performance (number of correctly predicted GCD), as the TB increases (Figure 2 Left).

For a modest training budget of 30 million, the models with the smallest DB (1 and 5 million, 1M and 5M-models henceforth) achieve the best performance (20 GCD vs 13 for all other DB). As TB increases, the 1M-models start overfitting, as shown by the increasing test losses in Figure 2 (Right), and their performance saturates at 21 correct GCD. The performance of the 5M models keeps improving to 36 GCD, for a TB of 150 million examples, then saturates around 38 GCD as the models overfit. For TB of 150 and 300 million examples, the best performing models are the 10M. As training proceeds, they are outperformed by the 25M models, which achieve the best performance for TB from 450 million to 1.05 billion examples (with the 50M-model a close second

---

[1]Power et al. (2022) also study modular *division*, equivalent to modular multiplication.

[2]For GCD and modular multiplication, input pairs are uniformly sampled integers from 1 to 1 million. In the unlimited data case, this gives rise to infrequent repetitions: over $\sim$ 1 billion input pairs, our largest data budget, no elements are repeated 3 or more times, and about 500 thousand are repeated twice.

at 1 billion). Throughout training, the models trained on small data budgets learn faster. However, past a certain TB, they overfit their training data, and their performance saturates.

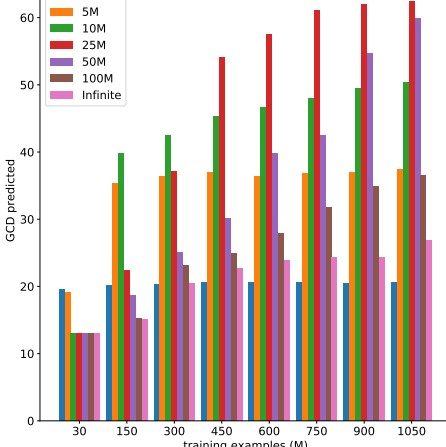 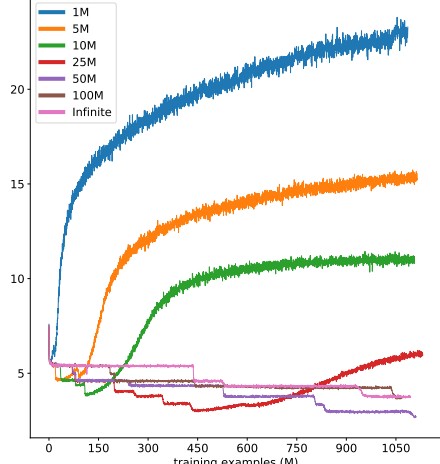

Figure 2: **GCD problem:** (Left) GCD accuracy for different data and training budgets (average of 5 models). (Right) Test loss of models as a function of training budget, for fixed data budgets.

**Note.** *Overfitting* is an overloaded term. In this paper, we define it by its empirical consequences: *a model overfits when its test loss starts increasing, while the train loss continues to decrease.* The relation between learning and overfitting is further studied in Appendix B.

Conversely, models trained with large or unlimited DB perform the worst. For a TB of one billion examples, the 25M-models predict 62 GCD on average, and the 50M-models 60. The 100M-models only predict 37 GCD and models trained on an unlimited data budget, where all training examples are seen only once, predict 27 GCD, *way worse* than models trained on 25M distinct examples, repeated 42 times on average. Summarizing, **smaller data budgets and more frequent repetition allow for faster learning, but also for much better performance.**

We observe a similar behavior for **modular multiplication**. For a TB of 600 million, we train 5 models for small DB, and 25 or 30 for larger DB, to zoom on this interesting region (Table 1). Models trained on an unlimited data budget perform at "chance level": they always predict 0 and achieve about 3% accuracy. Models trained on data budgets of 100 million examples fare little better, and models trained on 10 million examples or less overfit and do not learn.

Models trained on DB of 25M and 50M (for an average repetition of 24 and 12) achieve 40 and 60% accuracy and exhibit a different behavior. On this task, learning happens in sudden steps, separated by flat plateaus (see the empirical learning curves in Figure 7 in Appendix C), the two last plateaus corresponding to 51% and 99% accuracy. About 25% (7 out of 25) of 50M models achieve 99% accuracy (i.e. fully learn the task), and almost 90% (22/25) achieve 50% (i.e. one learning step away). On this task, **learning emerges through repetition**. Models trained on smaller data budgets can perform tasks that models trained from large or unlimited data budget cannot learn. To probe whether learning eventually occurs, we trained 19 models with unlimited and 100M DB on increased TB of 2 billion examples. None of the unlimited DB models could learn modular multiplication, but one 100M model out of 19 achieved 99% accuracy after 1B TB, and 5/19 after 2B.

|  | Data budget (millions) | | | | | | |
|---|---|---|---|---|---|---|---|
|  | 1 | 5 | 10 | 25 | 50 | 100 | unlimited |
| Average accuracy (%) | 1.6 | 3.8 | 4.4 | 40.4 | **59.5** | 5.4 | 3.0 |
| Number of models achieving 99% accuracy | 0/5 | 0/5 | 0/5 | 6/25 | **7/25** | 0/30 | 0/30 |
| Number of models achieving 50%+ accuracy | 0/5 | 0/5 | 0/5 | 13/25 | **22/25** | 0/30 | 0/30 |
| Number of models trained | 5 | 5 | 5 | 25 | 25 | 30 | 30 |

Table 1: **Multiplication modulo 67**. Accuracy of models trained on a budget of 600 million data points.

Finally, on the **eigenvalue problem**, Charton (2022) trained models with unlimited data budgets (DB≈TB) and observed that whereas 4-layer transformers can learn to compute the eigenvalues of $5 \times 5$ matrices, deeper models are required for larger problems: 6-layers for $8 \times 8$ matrices, 8 for $10 \times 10$ and 12 layers for $12 \times 12$ matrices. Even with large training budgets, 4-layer models where unable to learn the eigenvalues of 10 or 12 dimensional matrices.

In our experiments, we wanted to study whether smaller DB could *induce* small models to learn large problems. We trained 4-*layer* transformers to predict the eigenvalues of $10 \times 10$ matrices. We trained 5 models for each data budget of $1, 5, 10, 25, 50$ and 100M, and 5 for an unlimited DB (one pass over the training data), with TB up to 500 million. As expected, none of the models trained on unlimited DB did learn: all test accuracy remained close to 0. However, 4 of the 30 models trained on smaller DB achieved $99\%$ accuracy: 3 models trained on 50 million examples (repeated 10 times), and one model trained on 10 million (repeated 50 times). Scaling even further, to $12 \times 12$ matrices, still using 4-layer transformers, with a TB of 420 millions, 2 models (out of 35) begin learning: a 10M model achieved $21\%$ accuracy, and a 5M $3.5\%$. As in previous experiments, for a given training budget, smaller data budgets and repeated training examples prove beneficial, but on this task, **small datasets improve model scaling**. With small DB, problems that required 8-layer or 12-layer transformers can be learned by 4-layer models.

This first series of experiments clearly indicates that **repetition helps learning.** On three different tasks, for a fixed training budget, models trained on a small data budget, i.e. fewer distinct examples repeated several times, achieve much better performance than models trained from examples used only once or repeated a small number of times, as is customary in most recent works on language models (Muennighoff et al., 2023).

This phenomenon applies in different ways for different problems. On the GCD task, small DB allow for faster learning and higher accuracy. For modular multiplication, we observe emergence: a task inaccessible to models trained with large or unlimited DB is learned with small DB. Finally, for eigenvalues, small DB allow for better model scaling: tasks that normally require 8 or 12-layer transformers are learned by 4-layer models. But in all cases, the repetition achieved by small DB prove beneficial: **smaller data budgets with repetition can elicit "emergent learning"**.

## 5 TWO-SET TRAINING

The previous experiments demonstrate that for a fixed training budget, the optimal data budget is not the largest possible, as commonly practiced. On all three tasks, training from a set of distinct examples an order of magnitude smaller than the training budget, repeated many times, improves performance. We now turn to a different but related problem: how to best use a given data budget?

As we have seen, repeated examples help the model learn. Training from a small subset of the available data should therefore be beneficial, since it would increase repetition. However, models trained from very small datasets will eventually overfit their data, causing their accuracy to saturate. Yet, this can be prevented by increasing the size of the training set. To address these contradictory requirements – a small train set to increase repetition vs a large train set to avoid overfitting – we propose *two-set training*. We randomly split the training sample into a small set of examples that will be repeated many times during training, and a large set of examples that will be seen a few times only. By doing so, we hope that the small set fosters learning, while the large set prevents overfit.

Specifically, for a data budget of $N$ distinct examples, we randomly select $S < N$ examples that will form the repeated set – in practice, we shuffle the training set, and assign the $S$ first examples to the repeated set. During training, examples are selected from the repeated set with probability $p$, and from the $N - S$ others with probability $(1 - p)$. As a result, a model trained with a training budget of $T$ will see $pT$ examples from the repeated set, repeated $pT/S$ times on average, while the $N - S$ remaining examples will be repeated $(1 - p)T/(N - S)$ times on average. The repetition levels in both samples can be adjusted by choosing the values of $S$ and $p$. Note that the limiting cases $p = 0$ and $p = 1$ correspond to one-set training, with a data budget of $N - S$ and $S$ examples respectively.

On the **GCD problem**, models trained on a single set, with a data budget of 100 million examples and a training budget of 600 million, predict 27 GCD on average (Figure 2 (Left)). Experimenting with two-set training for different values of $S$ and $p$, we observe that models trained on a repeated set of $250,000$ examples or less, with a probability $p$ of 0.25 or 0.5, predict more than 62 GCD on

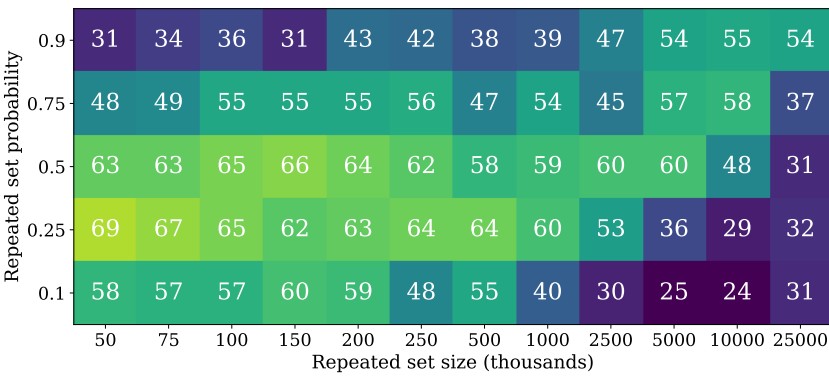

Figure 3: **Two-set training for the GCD problem:** Number of correctly predicted GCD as a function of $S$ and $p$. Each measurement is the average of 6 models. Data budget 100M, training budget 600M. Note the high performance for very small sets $S$ of sizes 50, 75, 100, 150 and 200 thousand, with $p = 0.25$ and $p = 0.5$.

average, a much better performance than their one-set counterparts. For $S = 50,000$ and $p = 0.25$, models predict 69 GCD on average, a better performance than the best models trained on a single set, with a larger training budget of 1 billion examples. For these parameters, the 50k examples in the small set are repeated $3,000$ times on average, and the rest of the training examples 4.5 times on average. On a 100M data budget, two-set training clearly outperforms single set training.

These results can be extended to unlimited training sets, by creating a fixed set of $S$ examples, selected with probability $p$, and generating (unlimited) random examples with probability $1 - p$. The best choices of $p$ and $S$ are roughly the same as with a DB of 100M (Figure 8 in Appendix C). In particular, with $p = 0.25$ and $S = 50,000$, two-set training on unlimited data achieves an average performance of 67 GCD on 6 models, a spectacular improvement over models trained on unlimited (single) datasets, which predict 25 GCD on average.

Therefore, for large and unlimited data budgets, frequent repetition of a tiny number of random examples, lost in a sea of single-use examples, unlocks surprising performance gains. Note the synergistic nature of this effect: training on the tiny sample alone (with large repetition), or one-set training on the same data budget, result in much lower performance than what two-set training provides by mixing them together (see also Appendix D.2: during training, mixing repeated and single-use examples into the same mini-batches is required for two-set training to happen).

We observe similar behavior for smaller data budgets. Figure 4 compares single and two-set training performance, for data budgets of 10, 25 and 50 million example, and training budgets up to 600M. For a given training budget, two-set training always achieves better performance than single-set training, and the benefit of two-set training increases as DB get larger. On this problem, two-set training accelerates learning. With large enough TB, single-set models sometimes catch up with the performance of their two-set counterparts with large enough TB (for 400M TB for 10M models, 600M for 25M, 1B TB for 50M models, see Figure 9, Appendix C). Still, most two-set models retain a marginal advantage[3] over models trained on a single set.

For **modular multiplication**, experiments with large and infinite data budget, for a training budget of 600M (Figure 5), indicate that larger repeated samples and smaller repetition, are needed, compared to GCD. With a DB of 100M, $S$ should be selected between 2.5 and 10 million examples, and $p$ between 0.25 and 0.5, for a small set repetition between 30 and 60 (vs 3000 for the GCD experiments). For unlimited DB, $S = 25M$ and $0.75 \leq p \leq 0.9$, a repetition between 18 and 22, seem optimal. Note also that in this problem, the choice of parameters $S$ and $p$ is more sentitive: only a few combinations allow for good performance (empirically, constant ratio between repetition on the small and large sample ($\frac{p(N-S)}{(1-p)S} \approx 10$).

However, with a careful choice of $p$ and $S$, two-set training achieves better performance than single set training for all data budgets from 25M to unlimited. Table 2 presents the proportion of models, trained on single and two sets, that learn to compute multiplication modulo 67, after a training

---

[3]and note that $S$ and $p$ might no longer be optimal for this larger training budget

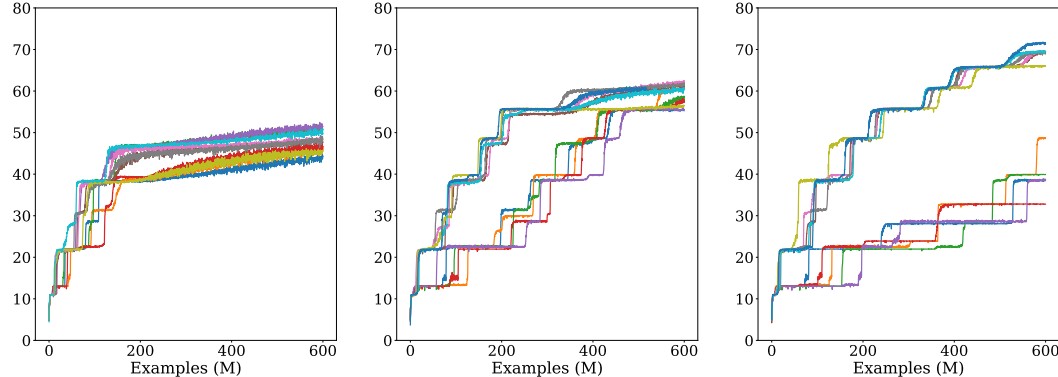

Figure 4: **Two-set versus single-set training for the GCD problem:** Number of correct GCD as a function of training budget (up to 600M) for data budgets of 10M (left), 25M (center), and 50M (right). Two-set training with $p = 0.25$ and $S = 50,000$ (top 6 curves) versus single-set training (lower 6 curves). See Figure 9 in Appendix C for extended TB with DB of 50M.

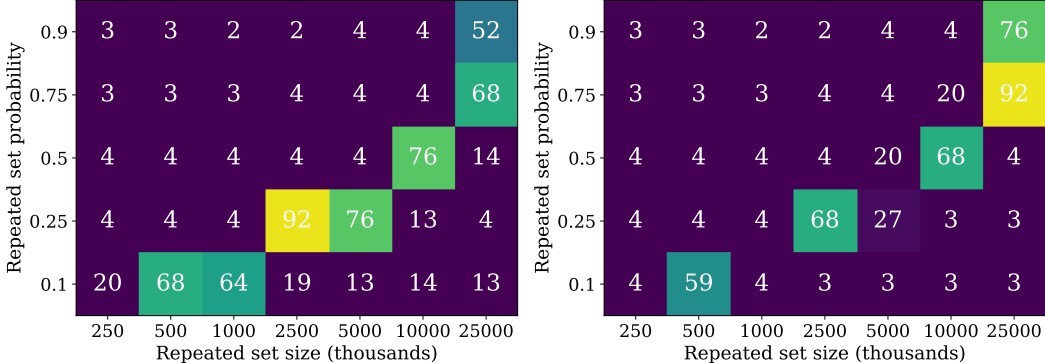

Figure 5: **Two-set training for Modular Multiplication:** Accuracy as a function of small set size $S$ and $p$, each averaged over 6 models. Data budget 100M (left) and unlimited (right), training budget 600M. Note: the bottom right of the left graph correspond to single-set 10M-models: for $p = 0.1$ and $S = 10$M, the small and large set are selected with the same probability.

budget of 600M. With two set training, 50 to $58\%$ of the models learn multiplication with $99\%$ accuracy. With single set training, 24 to $28\%$ learn for DB 25 and 50M, and none for larger DB. In these experiments, two-set training improves accuracy for all data budgets. However, its impact on learning speed (observed for GCD) is less conclusive (Table 7 in Appendix C).

| Data budget | p / S | Two sets | | Single set | |
|---|---|---|---|---|---|
| | | $> 50\%$ | $> 99\%$ | $> 50\%$ | $> 99\%$ |
| 25M | 0.1 / 1M | 50 | **50** | 52 | 24 |
| 50M | 0.25 / 2.5M | 90 | **50** | 88 | 28 |
| 100M | 0.5 / 10M | 88 | **54** | 0 | 0 |
| Unlimited | 0.25 / 2.5M | 92 | **58** | 0 | 0 |

Table 2: **Two-set training on modular multiplication.** Percentage of models (different random initializations) learning to compute modular multiplication with 50 and $99\%$ accuracy. Training budget: 600M. For DB 25M and 50M, 10 models with two-set training, and 25 with single set training. For DB 100M and unlimited, 26 models with two-set training, and 30 with single set training.

Finally, on the **eigenvalue problem** for $10 \times 10$ matrices, we train models with an unlimited data budget and a training budget of 500M. With these parameters, models trained on single sets do not learn (see Section 4), but two-set training achieves significant accuracy. For $p = 0.25$ we run 15 models each for 7 different sizes of $S$ between $15,000$ and $960,000$. 9 models out of 105 learn to predict with more than $60\%$ accuracy (Table 3). We see that injecting small, frequently repeated random subsets into the training data causes emergence of learning, where uniform repetition fails!

Note, again, the synergistic effect: neither training on the small set alone, nor training with unlimited data budget in one epoch would allow any learning at all - it is the combination of both that makes two-set training powerful!

| S (thousands) | Two sets | | | | | | | Single set |
|---|---|---|---|---|---|---|---|---|
| | 960 | 480 | 240 | 120 | 60 | 30 | 15 | |
| #models learning with 60% accuracy | 3/15 | 1/15 | 1/15 | 3/15 | 0/15 | 0/15 | 1/15 | 0/5 |

Table 3: **Two-set training on eigenvalues.** Number of models (different random initializations) learning to compute eigenvalues with over 60% accuracy. Training budget: 500M. Larger small sets achieve better results, single-set learning does not learn.

Overall, our experiments indicate that, for a given data budget, two-set training – repeating a small set of *randomly selected* during training – greatly improves model performance, either by accelerating learning (GCD), or increasing model accuracy (modular multiplication, eigenvalues). The size of the repeated set appears to be problem dependent: small for GCD, larger for modular multiplication and eigenvalues.

## 6 ABLATIONS AND VARIATIONS

In this section, we discuss possible improvements to two-set training. Detailed ablation results can be found in Appendix D.

**Curating the repeated sample.** In two-set training, repeated examples are randomly sampled from the available training data. We now experiment with a possible improvement: selecting the repeated examples. Perhaps what really matters is the repetition of a particular class of "informative" examples, as in curriculum learning. The GCD problem is particularly well suited for this type of investigation. Charton (2024) showed that increasing the proportion of small integers, or oversampling the tails of the distribution of GCD in the training set ($\text{Prob}(\text{GCD} = k) \sim \frac{1}{k^2}$), greatly improved model performance.

We experimented with three curation strategies for the repeated set: log-uniform and uniform distributions of operands and input, shown to be beneficial by Charton, "easy sets" featuring small input and outcomes, and "heavy tail sets" featuring large GCD. For each setting, we trained 5 models with four "good choices" of $S$ and $p$ (Table 4), a data budget of 100M and training budget of 600M.

| S / p | 50k / 0.25 | 150k / 0.25 | 150k / 0.5 | 500K / 0.5 |
|---|---|---|---|---|
| Log-uniform inputs | 55.9 | 59.4 | 57.9 | 62.0 |
| Uniform GCD | 55.9 | 54.5 | 41.9 | 54.9 |
| Log-uniform inputs and GCD | 62.2 | **71.7** | **66.5** | **72.6** |
| Small inputs (1-1000) | 61.2 | **67.5** | 62.6 | 62.9 |
| GCD 1- 10 | 59.9 | 63.8 | 55.8 | 62.3 |
| GCD products of 2 and 5 | 54.2 | 39.8 | 40.7 | 30.1 |
| All GCD but 1 | **65.4** | 63.7 | 56.7 | 58.1 |
| All GCD but 1,2,3 | **66.7** | 58.4 | 62.8 | 58.2 |
| All GCD but 1,2,3,4,5 | **66.5** | 60.6 | 64.9 | 56.3 |
| Baseline (two-set training from random examples) | **69.4** | 61.9 | **65.9** | 59.4 |

Table 4: **GCD problem: cherry-picking the repeated set**. Number of GCD predicted, average of 5 models (3 for baseline), training budget 600M. **bold**: more than 65 GCD predicted.

These strategies do not achieve better results than the baseline two-set training with a random repeated set. A slight improvement is observed when repeated samples are selected from a log-uniform input and GCD (for which Charton (2024) reports 91 correct GCD for single-set training). Overall, we find that repeated set curation has, at best, a marginal impact on performance. This is a counter-intuitive but significant result.

**Shifting the repeated sample.** In the GCD experiments, with $p = 0.25$ and $S = 50,000$, repeated examples are seen 3000 times for a training budget of 600M. Since this large repetition may lead

to overfit, we experimented with "shifting samples": replacing the repeated examples after a $k$ repetitions. In Appendix D.3, we experiment with $k$ from $10$ to $100$, and observe that this has no impact on model performance.

**Batching matters.** All models in this paper are trained on mini-batches of $64$ examples. In two-set training, batches mix examples from the repeated and the large set. We experimented with batches that only use samples from one set at a time. For instance, when training with $p = 0.25$, $25\%$ of batches would use repeated examples only. For both GCD and modular multiplication, we observe that models trained on batches from one sample only fail to learn. This indicates that mixing repeated and non-repeated examples is required for two-set training to happen (see also Appendix D.2).

**From two to many-set training.** Two-set training effectively makes the training sample non identically-distributed: examples from the repeated sample occur with a larger probability. We can generalize this method by introducing a probability distribution $P$ on the training examples, such that for any $i \leq N$, $P(i)$ is the probability that the $i$-th example is selected during training. In two-set training, $P$ is a step function distribution with two values: $p/S$ and $(1-p)/(N-S)$, we now replace it with a discrete exponential distribution $P(i) \sim \beta e^{-\beta i/N}$, with $\beta > 0$, suitably normalized. Table 5 presents the performance of models trained on the GCD problem with such "continuous" data distributions, indicating that our observations on two-set training generalize to such data sampling techniques. More details, and results on modular multiplication, can be found in Appendix D.4. These results suggests that our observations on two-set training can be extended to a wider class of methods, that use non-uniform sampling over a randomly ordered training set.

| $S_{\text{eff}}$ | 25k | 50k | 100k | 250k | 500k | 1M | 1.5M | 2M | 2.5M | 3M | 3.5M | 4M | 5M |
|---|---|---|---|---|---|---|---|---|---|---|---|---|---|
| $\beta$ | 1152 | 576 | 288 | 115 | 58 | 29 | 19 | 14 | 11.5 | 9.6 | 8.2 | 7.2 | 5.8 |
| GCD | 19 | 21 | 29 | 38 | 46 | 55 | 56 | 57 | 61 | 65 | 63 | 62 | 56 |

Table 5: **GCD for different exponential distributions.** Correctly predicted GCD, best of 5 models, trained on 600 million examples.

# 7 DISCUSSION

Our findings indicate that repetition, and possibly memorization, fosters learning. They suggest that models should be trained on datasets of repeated, but not necessarily curated examples, and that amplifying a randomly chosen subset of the training data may bring additional learning benefits. Two-set training is easy to implement, and applicable to a large variety of situations. Its extension to smooth distributions allows for finer control over repetition levels in the training sets. One feature of our tasks - which is the case in most reasoning tasks in AI4Math settings - is that they are deterministic : there is only one correct solution. For this community, our insights are of immediate relevance, as they give prescriptive advice on how to utilize training data (iterate rather than one-pass, use two-set training).

We can contemplate how our observations carry over to large language models (LLM) trained on natural data. An important factor is the presence of repetition in the training data. We believe that pre-training corpora – text scraped from the internet, public code repositories – feature many repeated examples (quotes, copied passages, duplicated functions), and that the phenomena we describe are already at work in LLMs during the pre-training stage. Fine-tuning corpora, on the other hand, are often curated and feature less repetition. We believe two-set training, and associated methods, may prove beneficial for fine-tuning LLMs.

Our observations on two-set training are thought-provoking and deserve further study. The fact that the repeated set can be chosen at random, and that curating repeated examples bring little to no improvement in performance suggest that what matters, here, is seeing the *exact same* example several times. The particulars of the example, its informational value, interest, whether it is typical or exceptional, seem to have little impact. This is all the more curious as, even in the two-set setting, repetition occurs at a very low frequency. In the two-set GCD experiments, repeated examples were seen 3000 times over a training budget of 600 million: once every 200,000 examples on average. The frequency is even lower for modular multiplication. Besides, the repeated examples are mixed with non-repeated examples into mini-batches, and our experiments indicate that this mixing is required for the two-set effect to appear. Still, this very infrequent repetition, and mini-batch mixing, brings a significant boost in model performance.

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

APPENDIX

# A ADDITIONAL DISCUSSION

Our empirical work studies the role of repetition in learning. In our two-set setting, a select subset of the data is presented to the learner at increased frequency. This is reminiscent of *continual learning*, where, to avoid catastrophic forgetting, certain examples from a previous task are repeated again. However, it is important to point out that continual learning refers to a setting with *distribution shift*: the learner is supposed to adjust to new distributions in the input - and to avoid catastrophic forgetting of earlier tasks. This is an extremely interesting research area - but it is not what we study. In our setting the distribution is always the same, both for repeated examples (Section 4) and for the new phenomenon of two-set training (Section 5). Both the more frequently sampled and the less frequent larger set come from exactly the same underlying distribution - which is what makes this effect so surprising.

Our work also resembles *curriculum learning*, where training data is presented in a particular order (usually from easy to hard). Our work is different from curriculum learning: We show that *randomly* selecting a small subset of the training data, and repeating them more often can significantly enhance performance or even overcome learning bottlenecks. We discover a synergistic effect: neither training on the small set alone, nor training with unlimited data budget in one epoch would allow any learning at all - it is the combination of both that makes two-set training powerful! The fact that the repeated set can be chosen at random, and that curating repeated examples brings no improvement in performance sets it aside from curriculum learning and suggest that what matters, here, is seeing the *exact same* example several times.

Can the two-step procedure be performed by showing only the repeated samples in a first learning phase, and then moving on to the more diverse samples in a second phase - thus placing it icloser to the realm of curriculum learning? We have performed variants of this experiment (see Appendix D.6) to provide a negative answer to this question.

Very few *theoretical* works have tried to study repeated examples in the learning process. Indeed, the fact that our observed two-set effect disappears when data is presented in mono-batches (either from the frequent set or the larger set), means that the mechanisms at play must involve optimization. Noteworthy are two recent works from Dandi et al. (2024) and Arnaboldi et al. (2024) in the special case of the the multi-index model for two-layer nets. There, it is shown that batches need to be seen more than once for beneficial symmetry breaking - and ultimately learning - to happen. Our mono-batch ablation shows that this cannot explain what we observe in our case, however. First, in our case, no two batches are the same, as data gets reshuffled. However, mono-batches are more likely to resemble each other when they come from the smaller, frequent set, compared to the multi-batch case where by design most examples in each batch will never be seen again (in the case of unlimited data at least). This means that our mono-batch case should resemble the scenario in Arnaboldi et al. (2024) more - yet it fails to show the observed effect.

This raises several tantalizing questions: how does the transformer "figure" that a given example, lost in a mini-batch, has been seen, several hundred thousand examples before? Our research suggests that there exists a qualitative difference between "déjà vu" and "jamais vu" examples – data points the model has already seen, or never seen. How do transformers, and perhaps other architectures, identify, and then process, "déjà vu" examples? To our knowledge, this aspect was overlooked in many prior works on model interpretation. We believe our findings point to a number of interesting questions about memorization in language models. This is an intriguing subject for further study.

# B LEARNING DYNAMICS AND OVERFITTING IN MATH TRANSFORMERS

To gain some understanding on the relation between repetition and overfitting, we delve deeper into the typical training dynamics in our mathematics problems with transformers. We study learning curves to shed light on the interplay between overfitting and relative size of data versus training budget. We focus on learning to compute the eigenvalues of $5 \times 5$ symmetric matrices (Charton, 2022) for illustrative purposes, but the observed dynamics are common to all our problems (e.g. see Figure 2 (Right)). Figure 6 illustrates training of 10 models on a data budget of $200,000$ samples,

with increasing training budget (up to 30 million) resulting in increased repetition. Learning curves exhibit a *step shape*, which gives rise to three phases:

- *Initial phase:* training and test loss decrease (up to TB of about 2M), accuracy remains low.
- *Learning phase:* training and test loss drop suddenly, accuracy increases steeply from a few percents to 90% (for the next 1-3M of TB). This phase is absent for those models that overfit too early (dark curves in Figure 6).
- *Saturation phase:* the model learns the remaining accuracy.

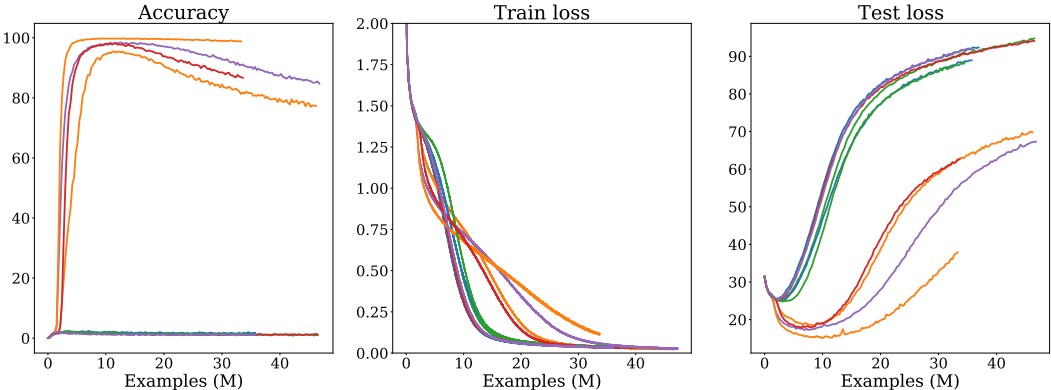

Figure 6: **Learning curves for eigenvalue computation of 5x5 matrices:** Accuracy, train and test loss, for 10 models trained on a data budget of $200,000$, as a function of training budget (TB). The curves represent different seeds. Note the *initial phase*, common to all curves, up to a sharp transition of test loss at $\sim$ 2M TB. At this point the dark curves begin to overfit (test loss increases) while the light curves undergo another drop in test loss that initiates the *learning phase*.

Recall that we say that *overfitting* occurs when the test loss starts increasing while training loss continues decreasing. Here we see that for *all* models there is an initial flattening of test loss after $\sim$ 2M training examples (about 10 repetitions of the data budget[4]). Then, some models start overfitting already during the initial phase (the 6 dark colored curves in Figure 6), and for those the learning phase never happens and accuracy plateaus at about 2%. On the other hand, for the other 4 models the learning phase begins before overfitting sets in (the pale colored curves in Figure 6), the task is learned in full (to over 95% accuracy), and overfitting is delayed until after that point. Eventually, these four models start to overfit at training budgets of about 10 million examples, and a slight drop in accuracy is observed in some models (but not all), after 15 million examples (75 epochs on the training set). We observe similar effects for different data budgets.

These experiments illustrate the relation between overfitting and learning. Once a model overfits, it stops learning, accuracy saturates, and eventually sometimes decreases. On the other hand, once a model trained on limited data starts learning, overfitting is delayed by many more epochs.

## C  ADDITIONAL FIGURES AND EXPERIMENTS FOR MODULAR MULTIPLICATION

Figure 7 provides learning curves (test error) for modular multiplication, illustrating step-like learning, which motivates us to use the number of models achieving $50 + \%$ resp. 99% accuracy as our performance metric.

Figures 8 and 9 as well as Tables 6 and 7 provide additional results for modular multiplication in the two-set setting.

---

[4]Our runs on a range of small data budgets (up to 250 thousand) show similar initial step shape of test loss at 10-12 repetitions.

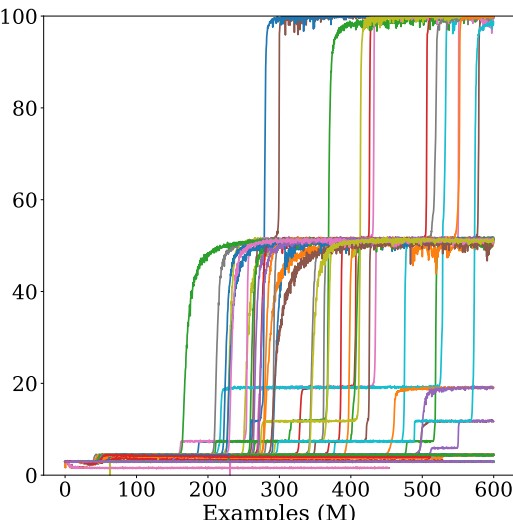

Figure 7: **Learning curves for modular multiplication:** Test accuracy for different model initializations. We see a clear step-like learning curve with a plateau just above $50\%$ accuracy before jumping to near perfect accuracy.

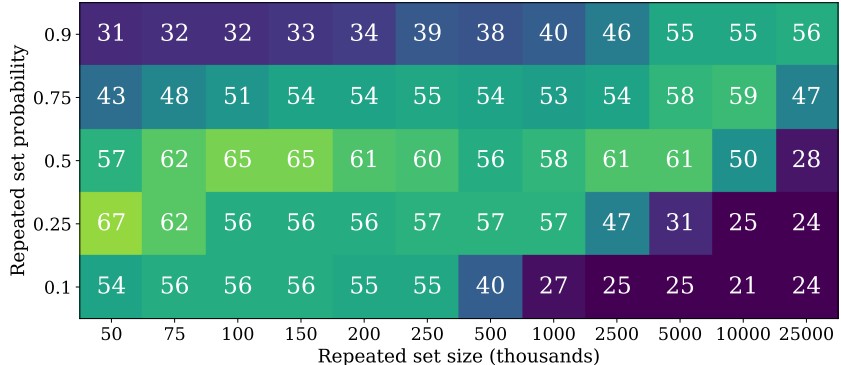

Figure 8: **Two-set training for the GCD problem for $\infty$-models:** Number of correctly predicted GCD as a function of small set size $S$ and $p$, each averaged over 6 models. Data budget *and* training budget equal 600M ($\infty$-models). Note the high performance for very small sets $S$ of sizes between 50 and 200 thousand, with $p = 0.25$ and $p = 0.5$ compared to "standard" training with the same data budget, predicting 25 GCD correctly (see Section 4 ).

# D    ABLATION RESULTS

## D.1    CURATING THE SMALL SAMPLE

In two-set training, the examples in the small set are chosen at random from the overall training set. In this section, we experiment with curating the small set, by *selecting* the examples that will be repeated during training. As in curriculum learning, selecting easier or more informative examples may help improve performance. Perhaps when increasing the frequency of our small random set, what really matters is the repetition of some particular examples, rather than all? The GCD problem is particularly well suited for this type of investigation, due to the inverse polynomial distribution of outcomes (Prob(GCD $= k) \sim \frac{1}{k^2}$). On this problem, we leverage the findings of Charton (2024), who observes that $\infty$-models trained from log-uniform distributions of inputs and/or outcomes (Prob(GCD $= k) \sim \frac{1}{k}$) learn better.

We experiment with four settings of $|S|$ and $p$, which correspond to the best results in our previous experiments (Section 5): $50,000$ and $150,000$ with $p = 0.25$ and $150,000$ and $500,000$ with

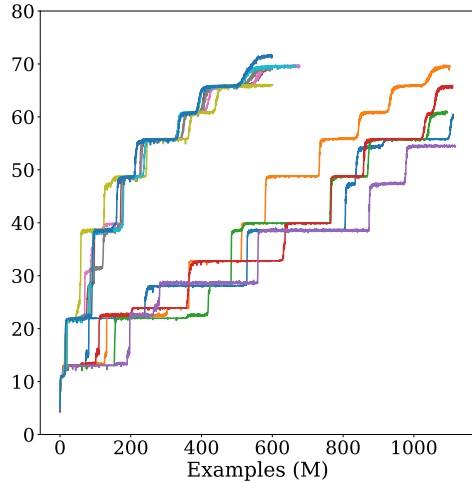

Figure 9: **Two-set versus single-set training for the GCD problem:** Number of correctly predicted (test) GCD as a function of training budget (up to 1B) and data budget of 50M Two-set training with $p = 0.25$ and $|S| = 50,000$ (top 6 curves) versus single-set training (lower 6 curves). With enough TB, single-set training achieves comparable performance with two-set training.

| $(p, S)$/ **Data budget** | 25M | | 50M | | 100M | | $\infty$ | |
|---|---|---|---|---|---|---|---|---|
| | $> 50\%$ | $99\%$ | $> 50\%$ | $99\%$ | $> 50\%$ | $99\%$ | $> 50\%$ | $99\%$ |
| $(0.1, 500K)$ | 2/10 | 1/10 | 6/10 | 3/10 | 20/26 | 10/26 | **25/26** | 8/26 |
| $(0.1, 1M)$ | **5/10** | **5/10** | 8/10 | 4/10 | 22/26 | 6/26 | 0/26 | 0/26 |
| $(0.25, 2.5M)$ | 2/10 | 1/10 | **9/10** | **5/10** | 20/26 | 9/26 | 24/26 | **15/26** |
| $(0.25, 5M)$ | 3/10 | 1/10 | 9/10 | 4/10 | **24/26** | 10/26 | 5/26 | 0/26 |
| $(0.5, 10M)$ | 3/10 | 3/10 | 8/10 | 5/10 | 23/26 | **14/26** | 23/26 | 12/26 |
| $(0.75, 25M)$ | - | - | - | - | 23/26 | 10/26 | 20/26 | 14/26 |
| Single set | 13/25 | 6/25 | 22/25 | 7/25 | 0/30 | 0/30 | 0/30 | 0/30 |

Table 6: **Two-set training on modular multiplication.** For a training budget of 600M we show the number of models (random initializations) that achieve $50 + \%$ and $90\%$ accuracy for several data budgets and sizes of the more frequent sets $S$, and probabilities $p$. The baseline of single-set traning from Section 4 is given in the last line. Similar results for training budgets of 300M and 450M are given in Table 7.

| | **Data budget** 25M | | | | | | **Data budget** 50M | | | | | |
|---|---|---|---|---|---|---|---|---|---|---|---|---|
| | $> 50\%$ | | | $99\%$ | | | $> 50\%$ | | | $99\%$ | | |
| | 300M | 450M | 600M | 300M | 450M | 600M | 300M | 450M | 600M | 300M | 450M | 600M |
| $(0.1, 500K)$ | 1 | 2 | 2 | 0 | 1 | 1 | 4 | 5 | 6 | 0 | 1 | 3 |
| $(0.1, 1M)$ | 1 | **5** | **5** | 0 | **3** | **5** | 3 | 6 | 8 | 0 | 1 | 4 |
| $(0.25, 2.5M)$ | 2 | 2 | 2 | 0 | 1 | 1 | 5 | **9** | **9** | 0 | 1 | **5** |
| $(0.25, 5M)$ | **3** | 3 | 3 | 0 | 0 | 1 | 4 | **9** | **9** | 0 | 1 | 4 |
| $(0.5, 10M)$ | 2 | 3 | 3 | 0 | 2 | 3 | **7** | 7 | 8 | 0 | **2** | **5** |
| Single set (/10) | 3.6 | 4.8 | 5.2 | 0.4 | 1.2 | 2.4 | 2.4 | 7.6 | 8.8 | 0 | 0.8 | 2.8 |
| Single set (/25) | 9/25 | 12/25 | 13/25 | 1/25 | 3/25 | 6/25 | 6/25 | 19/25 | 22/25 | 0/25 | 2/25 | 7/25 |

Table 7: **Two-set training on modular multiplication.** For training budgets of 300M, 450M and 600M we show the number of models out of 10 (random initializations) that achieve $50 + \%$ and $90\%$ accuracy for data budgets 25M and 50M, and sizes of the more frequent sets $S$, and probabilities $p$. The baseline of single-set training is given in the last line, out of 25 models. The next to last line renormalizes this to out of 10.

$p = 0.5$, for a data budget of 100 million and training budget of 600M. For every setting, we train 5 models with the following three choices for $S$: log-uniform inputs, uniform GCD or both log-uniform inputs and GCD. We use two-set training with a random small set $S$ as our baseline. Table 8 shows that the performance of models using log-uniform inputs, or uniform GCD, is slightly lower than the baseline. Models trained on log-uniform inputs and GCD achieve slightly better per-

formance, but we note that models trained on the small set distribution only ($p = 1$) would predict 91 GCD. On these three distributions, curating the small set proves disappointing.

To summarize our first observations here: Prior work (Charton, 2024) on solving GCD with transformers shows that a form of curriculum learning/data curation, namely training on log uniform operands, does improve performance. The best performance of training with log-uniform operands is about the same as our proposed two sample training. One cannot improve performance by mixing the two, i.e. by curating the small, more frequently seen subset, to have log uniform operands.

In curriculum learning fashion, we also experiment with small sets $S$ of a few "easier cases": small inputs (from 1 to 1000), GCD that are products of 2 and 5, the easiest to learn in base 1000 (Charton, 2024), and GCD between 1 and 10 (the most common outcomes). We observe that while models trained with small inputs in $S$ perform on par with the baseline, models trained on "easy GCD" perform slightly worse.

Finally, inspired by arguments that rare tail outcomes might require particular attention for learning (Dohmatob et al., 2024), we experiment with small sets composed of examples from the tail of the training distribution, namely, large GCD. Charton (2024) observes that these are both harder to learn, and less common in the training set. Specifically, we create $S$ with examples with GCD larger than $k$ (for $k$ ranging from 1 to 5). While experiments achieve the best accuracies compared to the other curation schemes we proposed, and values of $k$ equal to 2 and 3 train slightly faster, they remain a little below the baseline both in accuracy and learning speed.

| | 50k / 0.25 | 150k / 0.25 | 150k / 0.5 | 500K / 0.5 | Training budget for 60 GCD (M) |
|---|---|---|---|---|---|
| Log-uniform inputs | 55.9 | 59.4 | 57.9 | 62.0 | 332 |
| Uniform GCD | 55.9 | 54.5 | 41.9 | 54.9 | - |
| Log-uniform inputs and GCD | 62.2 | **71.7** | **66.5** | **72.6** | 88 |
| Small inputs (1-1000) | 61.2 | **67.5** | 62.6 | 62.9 | 247 |
| GCD 1- 10 | 59.9 | 63.8 | 55.8 | 62.3 | 401 |
| GCD products of 2 and 5 | 54.2 | 39.8 | 40.7 | 30.1 | 548 |
| All GCD but 1 | **65.4** | 63.7 | 56.7 | 58.1 | 405 |
| All GCD but 1,2 | **66.8** | 60.0 | 62.8 | 56.9 | 326 |
| All GCD but 1,2,3 | **66.7** | 58.4 | 62.8 | 58.2 | 327 |
| All GCD but 1,2,3,4 | **65.5** | 60.3 | 62.8 | 56.9 | 379 |
| All GCD but 1,2,3,4,5 | **66.5** | 60.6 | 64.9 | 56.3 | 376 |
| GCD product of 2, 3, and 5 | **66.1** | 59.4 | 59.8 | 47.3 | 359 |
| Prime GCD | 64.9 | 62.5 | 58.8 | 64.7 | 422 |
| GCD divisible by primes $\geq 11$ | 60.1 | 54.4 | 35.7 | 42.7 | 569 |
| Baseline (two-set training) | **69.4** | 61.9 | **65.9** | 59.4 | 373 |

Table 8: **GCD problem: cherry-picking the small set**. (Left) Number of (test) GCD predicted for training budget of 600 million examples, average of 5 models (3 models for baseline). **bold**: more than 65 GCD predicted. (Right) Training budget needed to predict 60 GCD, fastest of 20 models (of 12 models for baseline).

Overall, these experiments suggest that in two-set training, random selection of the small set may be optimal. Selecting a small set of easy cases (GCD multiple of 2 and 5), and examples that are known to help training (log-uniform inputs) does not help, and limiting the small set to edge cases from the tail of the outcome distribution brings no improvement to performance. This is a counter-intuitive, but significant result.

### D.2 BATCHING IN TWO-SET TRAINING: MIXED BATCHES ARE NEEDED

In all experiments, during training, the model computes gradients over minibatches of 64 examples. In two-set training, minibatches mix examples from the small and large set. We experimented with using "mono-batches" that use samples from one set at a time. For instance, when training with $p = 0.25$, 25% of minibatches would use examples from the small set (of size $S$) only, and 75% would only use those from its complement.

On the **GCD problem**, we rerun the most successful two-set experiments (Section 5) with "mono-batches" for $S = 50K$, $100K$ and $250K$, and $p = 0.25$ and $0.5$. For training budgets of 600M and data budget of 100M examples, the models trained on mixed batches predicted 62 to 69 GCD (Section 5). With "mono-batches", the number of correctly predicted GCD never rises above 15. For **modular multiplication**, we experimented with the following $(S, p)$ pairs ($S$ in millions): $(0.5, 0.1)$, $(2.5, 0.25)$ and $(10, 0.5)$ with data budget 100M and training budget 600M. With these settings, mixed-batch models achieve an average accuracy of 67% or more (Section 5). With "mono-batches", none of the models manages to learn (accuracy around 4%). This indicates that **mixed batching of samples from each of the two sets plays a central role for the two-set effect**.

## D.3 SHIFTING THE SMALL SET

In these experiments, we study, in two-set training, the possible impact of overfitting on the small set, by refreshing the small set with fresh examples periodically. This mimics certain aspects of curriculum learning, where the training set is changed over time. On the GCD experiments, with a data budget of 100 million, a training budget of 600 million, we shift the small set as training proceeds, so that examples in the small set are seen $k$ times on average. At the beginning of training, the small set is the $S$ first elements in the train set. After training on $kS/p$ examples, examples in the small set have been seen $k$ times, and the small set is shifted to elements $S + 1$ to $2S$ of the training set.

Table 9 provides performances for two-set training with shift, for different values of $p$, $S$ and $k$, for a data budget of 100 million, and a training budget of 600 million. It is interesting to note that shifting brings no improvement to 2-set training.

| $S$ | 250,000 | | | | 500,000 | | | | 1,000,000 | | | |
|---|---|---|---|---|---|---|---|---|---|---|---|---|
| k | 10 | 25 | 50 | 100 | 10 | 25 | 50 | 100 | 10 | 25 | 50 | 100 |
| $p = 1.0$ | 37 | 22 | 21 | 22 | 37 | 38 | 30 | 31 | 55 | 45 | 37 | 30 |
| $p = 0.9$ | 47 | 38 | 38 | 38 | 55 | 47 | 43 | 39 | 55 | 48 | 47 | 47 |
| $p = 0.75$ | 56 | 38 | 54 | 48 | 56 | 55 | 49 | 55 | 60 | 56 | 55 | 56 |
| $p = 0.5$ | 61 | 56 | 56 | 58 | 61 | 60 | 56 | 58 | 64 | 63 | 63 | 61 |
| $p = 0.25$ | 56 | 62 | 61 | 63 | 49 | 63 | 63 | 61 | 49 | 63 | 62 | 63 |

Table 9: **Shifted two-set training.** GCD predicted, average of 3 models, trained on a budget of 600 millions, and a data budget of 100 million, for different values of S, p and k.

## D.4 FROM TWO-SET TO MANY-SET TRAINING

Two-set training with a small randomly selected subset $S$ amounts to assigning different probabilities to elements in the training set. For a randomly shuffled training set of size $N$, two-set training amounts to selecting the first $S$ elements with probability $p/S$ (with replacement) and the $N - S$ last with probability $(1 - p)/(N - S)$, a step-function distribution over $\{1, \ldots, N\}$. We now generalize this approach by introducing a probability law $P$ such that $P(i)$ is the probability of selecting the $i$-th example in the training set. Our motivation is to obtain a smooth, possibly more principled, distribution than the step-function induced by the two-set approach. Pragmatically, a one-parameter family of smooth distributions eliminates the need to tune both $S$ and $p$. Lastly, we can study whether a smooth decay in frequency might be even more beneficial than a non-continuous two-set partition.

In this section, we consider a discrete exponential distribution:

$$P(i) \sim \beta e^{-\beta i/N},$$

with $\beta > 0$, suitably normalized[5]. If $\beta$ tends to 0, $P$ tends to the uniform distribution, and implements the single-set strategy of Section 4. As $\beta$ becomes large, a small fraction of the full training

---

[5]The normalization factor is $(1 - e^{-\beta})^{-1}$. In our calculations we will approximate it by 1 to simplify computing $S_{\text{eff}}$. For the range of $\beta$ we consider, the resulting approximation error is negligible. In general, for fixed $p$, to compute the size of the set $S(p)$ of first elements that carry probability mass $p$, we can use $\beta \approx -\ln(1 - p)N/|S(p)|$.

set is sampled (99% of the probability mass lies on the $4.6N/\beta$ first elements, 99.99% on the first $9.2N/\beta$). For intermediate values of $\beta$, the model oversamples the first elements in the training set, and undersamples the last: we have a continuous version of two-set training. To allow for comparison with two-set training, we define $S_{\text{eff}}$ such that the first $S_{\text{eff}}$ examples in the training set jointly are sampled with probability 25%. In this setting, 10% of the probability mass is on the $0.37S_{\text{eff}}$ first training examples, and 99% on the first $16S_{\text{eff}}$.

For GCD, we experiment with values of $\beta$ ranging from 5.8 to 1152 ($S_{\text{eff}}$ from 25,000 to 5 million)[6]. Table 5 shows that for our training budget of 600 million examples, the best model ($S_{\text{eff}} = 3\text{M}$) predicts 65 correct GCD, slightly less than what was achieved with two-set training (Section 5).

For modular multiplication, we need lower $\beta$ (i.e larger $S_{\text{eff}}$) for our training budget of 600M. We report the number of models (out of 25 for each setting) that learn to accuracy above 50% and 95% respectively (Table 10). Again we see that these results are comparable to two-set training (Section 5).

| $S_{\text{eff}}$ | 2.5M | 5M | 6M | 8M | 10M | 12M | 14M |
|---|---|---|---|---|---|---|---|
| $\beta$ | 11.5 | 5.8 | 4.8 | 3.6 | 2.9 | 2.4 | 2.1 |
| # Models with 95% accuracy | 2 | 9 | 11 | 13 | 7 | 4 | 3 |
| # Models with 50% accuracy | 4 | 16 | 25 | 22 | 17 | 13 | 6 |

Table 10: **Modular multiplication with different exponential distributions.** 25 models trained on 600 million examples.

We conclude that the benefits observed in two-set training do not pertain to the specific two-set partition of the training set; rather, it seems that the core of the effect lies in the non-uniform sampling frequency distribution over the (randomly ordered) training set, with a range of frequencies.

### D.5 VARYING THE OPTIMIZER

Some effects observed in deep learning depend on the optimizer, with grokking being a prominent example (Power et al., 2022). Here we provide experimental evidence to show that our findings hold for a variety of optimizers and are thus *robust* and *universal*. We rerun models used for the GCD problem with different optimizers. Specifically, we trained models to predict GCD, with a training budget of 600 million examples, single and two-set training (with $|S| = 50,000$ and $p = 0.25$), and data budgets of 25 million, 50 million and unlimited. We considered four optimizer settings:

- Adam without dropout or weight decay,
- Adam with weight decay 0.01,
- Adam with dropout (0.1) in the feed-forward networks of the transformer,
- AdamW with weight decay 0.01.

Table 11 presents the best performance of 5 models for each configuration. On average, dropout has an adverse effect on learning, but there is no clear benefit of using weight decay, or AdamW over Adam. Importantly, the separation in performance between single-epoch unlimited training, training on smaller data budgets with more repetitions and two-set training persists across optimizers: the effects we present are robust.

### D.6 TWO-SET AS CURRICULUM LEARNING?

Here we study whether the two-step procedure can be performed by showing only the repeated samples in a first learning phase, and then moving on to the more diverse samples in a second phase. We have run this experiment, for both the GCD and modular multiplication task. We first train the model on a repeated set (or 50K and 2.5M examples respectively), for a fixed number of epochs, then switch to a larger set (of 100M examples). For each task, we ran 50 experiments: 10 model initializations, and 5 levels of repetition. All models overfit very rapidly, causing lower performance than previously observed (8 correct GCD predicted, vs 13 in previous experiments). Once the model

---

[6]Note that for these values of $\beta$ the distinction between DB 100M and unlimited DB becomes essentially meaningless, as the tails of the training set are sampled exceedingly rarely.

|  | One-set | | | Two-set | | |
|---|---|---|---|---|---|---|
|  | Unlimited | 50M | 25M | Unlimited | 50M | 25M |
| Adam | 28 | 49 | 61 | 70 | 72 | 63 |
| Adam wd=0.01 | 30 | 56 | 61 | 70 | 70 | 66 |
| AdamW wd=0.01 | 29 | 50 | 58 | 69 | 72 | 67 |
| Adam dropout=0.1 | 24 | 40 | 49 | 66 | 66 | 66 |

Table 11: **Modular multiplication with different optimizers.** Correctly predicted GCD of the best (of 5) models for various optimizers. The effects we observe are robust under change of optimizer, with a very small degradation for dropout for both the unlimited (single-epoch) and limited DB.

switches to the large training set, catastrophic forgetting sets in, and the model seems to learn from scratch. We note that this second phase is slightly delayed: overfitting seems to result in a bad initialization of weights.

We believe these results are also explained by our observation on mini-batches (Section 6 and Appendix D.2). If repeated and non repeated examples are presented separately, i.e. not mixed into the same mini-batches, the benefits of two-sample training disappear.

