# OpenReview forum: "Emergent properties with repeated examples"
_ICLR.cc/2025/Conference — ICLR 2025 Conference Withdrawn Submission_

### Official Review · Reviewer_bpnp · 2024-10-28

**Soundness:** 2
**Presentation:** 3
**Contribution:** 1
**Rating:** 5
**Confidence:** 4

**Summary:**

Modern language models are typically trained with only one or at most a few epochs. This paper studies the potential benefit of training for more than a few epochs. With experiments on three synthetic problems (GCD, modular multiplication and matrix eigenvalues), the paper shows that for certain compute budget (or "training budget" as termed in the paper), repeating the same data many times can actually be better than using completely fresh data (i.e. 1 epoch or "online" training). Based on this intuition, this paper further proposed a "two-set training" paradigm where only a subset of training examples is repeated, to get the benefit of repeated training while also mitigating the potential overfitting issue.

**Strengths:**

The paper presented the idea clearly and conducted controlled experiments to study the question. It also proposed new training paradigms based on the observations that could potentially help improving model training.

**Weaknesses:**

1. This paper is making very big claims (e.g. of "emergent properties" or "emergent learning") based on very synthetic settings. While the models used are transformer models, they are tasked with specialized tokenizers for numbers and trained to perform a single synthetic task (e.g. given two numbers, output the greatest common divisor of the two numbers). In other words, those "language models" do not have any language capabilities. I think the paper could be better either (A) frame this as a general machine learning problem and have comparison studies covering many different neural network architectures, or (B) focus on LM but have additional experiments on more realistic settings. For example, in terms of relevancy to LMs, I think a study of fine-tuning with a proper LM would be more "transferable" to our understanding of how LM learning works than this synthetic pre-training setting.

2. There was the classical decomposition of test accuracy to training accuracy + generalization gap. It seems especially relevant when the paper talks about training with or without repetitions on difficult math / arithmetic tasks. Many of the observations might be more intuitively explained by undertraining --- i.e. the training accuracy itself is already quite bad as the model struggle to fit the problem (with small data repetitions). Adding studies from this angle could potentially make the paper more clear, but in the meantime, the observations would be less surprising if it is mostly explained by such classical decomposition.

3. The paper proposed two-set training algorithm, but there is no comparison to any previous curriculum learning baseline algorithms.

4. It is not clear if the experiments are comprehensive enough to support some of the big claims. For example, the paper talks about "emergence", *a task inaccessible to models trained with large or unlimited DB is learned with small DB*. The conclusion may change depending on the model architecture or even model sizes, and if may even be unclear if each setting allows to choose their own optimal hyperparameters. This paper is making those conclusions based on experiments with a single transformer model with fixed number of parameters and a fixed set of hyperparameters. For example, with repeated examples, later training steps would have potentially smaller updates because the gradients are smaller on seen examples. If this is benefiting the learning, could a similar effects be achieved by a better learning rate decaying scheme?

**Questions:**

Please see the "Weakness" section above.

---

**Post Rebuttal**: Thanks the authors for the response. I raised my rating, but I'm still not super convinced about the setting as well as concrete evidences in the experiments of this paper excluding the undertraining as a potential confounding factor.

---

> ### Author Response · Authors · 2024-11-19
> **Author's response (1/2)**
>
> We thank the reviewer for their time and comments and your appreciation of the clarity of presentation and the training paradigm.
>
>
> >Weaknesses:
> This paper is making very big claims (e.g. of "emergent properties" or "emergent learning") based on very synthetic settings. [...]
>
> As noted in the beginning of our response to reviewer **knen**, we believe that we have aimed not to make general claims: all the claims on emergence or learning with repetition are associated to specific tasks. Please also kindly see our General Response and the discussion on "emergence" with reviewer yjc8, which might help to ground the discussion.
> This said, we believe our observations on modular multiplication are a clear case of emergence: without repetition, nothing is learned. With repetition, the model learns to performn the task. The GCD experiment indicates that this is not limited to one task. We make no claim about the applicability to very different fields (e.g. vision) or architectures (MLP or CNN); we are focusing on transformer architectures as they are used on contemporary NLP. On the other hand, the tasks considered are typical of the reasoning tasks which are a major field of current research on LLM. As we argue in our response to reviewer uAYC:
> Recent work on compute-optimal language models [Hoffman et al. 22] shows that many previously trained large language models could have attained better performance for a given compute budget by training a smaller model on more data. Most prior large language models have been trained for a single epoch [Komatsuzaki et al.'19 *"One epoch is all you need"*] and some work explicitly advocates against reusing data [Hernandez et al. 22]. [Muennighoff et al. NeurIPS 2023 Best Paper] undertake an extensive study of multi-epoch training for LLMs on natural data. They find that, while models trained for a single epoch consistently have the best validation loss per compute, differences tend to be insignificant among models trained for up to 4 epochs and do not lead to differences in downstream task performance (but surely not to any improvements). There is thus a deep-rooted belief in the transformer community that single-epoch training yields the best performance, and only when data is constrained some studies show that for a limited number of epochs (up to 4) can still yield some benefits (though not comparable to training on more data). Multi-epoch training is viewed as a poor proxy in attempts to attain the performance of single-epoch training if data was abundant. Repetition of training sets for transformers is viewed as a bug, not a feature, only to be employed when data is scarce.
>
> In controlled transformer experiments we challenge the *single-epoch paradigm*: not only is repeated training on smaller data budgets powerful competition to single epoch-training (for the same number of training steps); in several cases repeated training on a smaller set allows to *unlock* capacities that are unattainable with single epoch training on a much larger dataset. In some cases, increased repetition of a smaller set leads to emergent phenomena of learning.
>
>
>
> >There was the classical decomposition of test accuracy to training accuracy + generalization gap. [...] Many of the observations might be more intuitively explained by undertraining [....]
>
> We respectfully disagree. See Section 3 and especially Figure 1: models trained on very large datasets do not learn, even after more than a billion examples. Models trained on small datasets learn better, for the same training/compute budget. If undertraining was the problem it would affect small and large training sets. The generalization bounds you refer to also tend to improve with training data set size (though as we know, they tend to be vacuous for DL models and transformers anyway). Please let us know if we might have misunderstood your question.

---

> > ### Author Response · Authors · 2024-11-19
> >
> > >The paper proposed two-set training algorithm, but there is no comparison to any previous curriculum learning baseline algorithms.
> >
> > We are surprised by your question: We were very puzzled by our two-set training results, and have thus precisely provided a wide range of curriculum learning ablations (initially, we thought that the effect might be due to seeing tailed parts of the distribution several times - but our curriculum/data curation studies find this is not at all the case). In the beginning of Section 6 and Appendix D.1, we show that curating the repeated sample does not improve two-sample training performance. To be specific, we show:
> > 1. Prior work [Charton 2024] on solving GCD with transformers shows that a form of curriculum learning/data curation, namely training on log uniform operands, does improve performance
> > 2. The best performance of training with log-uniform operands is about the same as our proposed two sample training
> > 3. One cannot improve performance by mixing the two, i.e. by curating the small, more frequently seen subset, to have log uniform operands
> >
> > We have no comparison baseline for modular multiplication nor eigenvalue computation, which to our knowledge were not studied in prior work on curriculum learning.
> >
> > >It is not clear if the experiments are comprehensive enough to support some of the big claims. For example, the paper talks about "emergence", a task inaccessible to models trained with large or unlimited DB is learned with small DB. The conclusion may change depending on the model architecture or even model sizes, [...]
> >
> >
> > We would like to respectfully point out that we believe that we have aimed not to make general claims: all the claims on emergence or learning with repetition are associated to specific tasks. Please also kindly see our General Response on the synthetic data setting and the immediate applicability of our results to an entire branch of transformer-based AI4Math; as well as the discussion on "emergence" with reviewer yjc8, which might help to ground the discussion. In each setting, we have optimized the hyperparameters to constitute a fair comparison. We hope to have provided a broad rnage of settings to study the effects we discover, for each task. Note also our ablation in Appendix D.5 that shows the persistence of our observed results with varying optimizers (which is somewhat unlike the grokking phenomenon - also studied with synthetic data - which is much more suceptible to the choice of optimizer). If you feel these results warrant more focus in the main body of the paper, we will aim to summarize them there as well. We have also tested that learning rate schedules do not change our results - while we did not judge this as important enough to include, we are happy to do so if you feel this is important.
> >
> > We hope that our responses elucidate our framing and the general context of our work and hope that you would consider raising your score.

---

> > ### Comment · Reviewer_bpnp · 2024-11-22
> > **Thanks for the rebuttal**
> >
> > Thanks the authors for the rebuttal.
> >
> > > Synthetic data; challenging single-epoch paradigm
> >
> > I have no complain about synthetic data, and I believe synthetic data and controlled experiments are very important. But different kind of synthetic data serves different purposes. The authors cited many previous work using synthetic data in the general response above, but note, for example, in the line of work by Zeyuan Allen-Zhu, the experiments are carried out on synthetic problems in natural language, which look the same to other real datasets that a standard LLM training pipeline would use. On the other hand, this paper uses specialized problem format and specialized tokenizer that does not resemble the data consumed by typical LLM. Of course, there is nothing wrong in using this kind of synthetic data, but it is not very related to the LLM regime. Therefore, I find it quite confusing that this paper is "challenging the single-epoch paradigm" (i.e. the LLM training paradigm) with those experiments. Note the model architecture itself does not represents the "LLM regime or single-epoch paradigm", as Vision Transformers are apparently trained for many epochs.
> >
> > In the general response, the authors suggested that minimizing a cross-entropy loss is not different from other "letter games", such as code refactoring, text summarizing, or various reasoning tasks. I agree that going up to an abstract level, everything is just a machine learning problem, and it is unclear if there is anything fundamentally different between LLM training and a 1D curve fitting problem. However, if one takes this view, then there is no reason to consider "one epoch training" a paradigm to challenge as most standard ML problems are not trained for only one epoch.
> >
> > In summary, I find a mismatch between the experiments and the claim: if the focus is LLM regime, I do not think the synthetic problems are close enough to actual language modeling data to provide insights to actual LLM training; if the focus is standard ML (for which case the synthetic problems could be convincing), then "one epoch training" is not a standard practice to be challenged.
> >
> > > undertraining; See Section 3 and especially Figure 1
> >
> > Apologize that I missed Figure 1, which did not seem to be mentioned anywhere in the text. But I don't think it answers my question after checking it.
> >
> > The authors argue that undertraining was not the issue because it would affect small and large training sets. I'm not sure if undertraining can be easily captured by the size of the training set. Since we may not be on the same page, let me explain my concern again: the decomposition of training accuracy + generalization gap could sometimes be very helpful for understanding why poor performance happens. In the case of undertraining, the (vacuous or not) generalization gap is usually not the main concern, because the training accuracy itself is bad. It could be that the model does not have enough capacity, or the training problems are too complicated to fit or the optimization algorithm is not suited, etc. In any case, bad training accuracy is an *optimization* problem, not a *learning* problem. What I was asking was to provide the training accuracy (together with the test accuracy) when comparing different setups, so that we can get a sense when the model does not "learn" on a billion examples, whether or not it is due to optimization difficulties.

---

> > > ### Author Response · Authors · 2024-11-25
> > > **Reply to your comments**
> > >
> > > **Clarifying our claim and the single epoch paradigm**
> > >
> > > Thank you for your reply. We consider small transformer architectures (4 layers, 512 dimensions), trained from scratch and with supervision, from synthetic data with a controlled distribution. This is a different setting from LLMs: larger models, pre-trained unsupervisedly, on huge dataset with no control over the data distribution. We do believe it would be interesting to extend our work to LLMs, but as we state in the conclusion, a lot of additional work is required to measure and control repetition in (scraped) pre-training data. So far, our conclusions hold for small transformers, such as the one routinely used in maths and science.
> > >
> > > The tokenizer we use is in fact very close to those typically used by LLMs. In the text data fed into LLMs, integers are usually "cut" into subsequences of 2 or 3 digits: e.g. 102345 is tokenized either as 10,  345, or as 10, 23, 45. The base 1000 tokenizer we use represents this number as + 102 345. Shorter sequences, like 1234, used to be tokenized as  123 4, but several recent papers suggested right-aligning the integer before tokenization, which results in the tokenization of 1234 as 1, 234: exactly the same as the base 1000 method we propose.
> > >
> > > Because of this, the arithmetic problems considered in this paper (GCD and modular multiplication) are actually encoded in a very similar way to how they would be presented to LLMs.
> > >
> > > Let us clarify what we mean by "letter games" and the use of cross-entropy. When solving numerical problems, it is very tempting to use mean squared error, or similar numerical loss. By "telling" the model that predicting a 2 as a 3 is better than predicting it as a 5, MSE loss introduces a problem-specific inductive bias about model inputs and outputs. Somehow, the use of MSE tells the model we are doing maths, and that there is a natural ordering of tokens. Cross-enttropy, on the other hand, carries no such bias: the digits in numbers could be all renamed (into red, blue, green, or plum, tomato and cherry) without impacting our conclusions. This is what we mean by a "letter game": the model is unaware of the underlying mathematics, and the tokens used to represent numbers are meaningless. We believe the use of cross-entropy helps counter the possible criticism that our results are specific to problems of mathematics.
> > >
> > > On the single epoch paradigm, we disagree that it would be specific to LLMs. The idea that larger datasets are better is common across all areas of machine learning. Some models indeed use multi-epoch training, but in most cases, it is *because the data is sparse*, not a deliberate choice by model trainers. We claim, on the other hand, that even in situations where they can generate as many data as they want, they should abstain to, and work from repeated examples.
> > >
> > >
> > > **About undertraining and generalization gaps**
> > >
> > > We believe we can rule out the possibility of undertraining in the case of the GCD which was studied in detail in prior work (Charton 2024). Charton indicates that the performance levels observed hold over a broad range of hyper-parameters (model size and architecture, optimizer settings). Also, we confirm his observation that models learn a set of deterministic rules, which "explain" their predictions, and apply the same to "seen" training data, and "unseen" test data. More generally, in all experiments, we observe no difference in accuracy on the training and test set, which seems to rule out the existence of a generalization gap.
> > >
> > > Finally, we took care to run models with very large compute budgets (1 billion examples for GCD, 2 billion for modular arithmetic, see Section 4) and saw no change in our conclusion about "unlimited" training sets. Whereas we agree that undertraining would weaken our conclusions, we see very little sign of it in the experiments we presented.
> > >
> > > We hope these additional clarifications might be helpful to decide whether you would reconsider your current score (please also kindly see the General Response and the new uploaded version incorporation your and other reviewers' feedback.) Thank you!

---

### Official Review · Reviewer_uAYC · 2024-10-31

**Soundness:** 2
**Presentation:** 3
**Contribution:** 2
**Rating:** 5
**Confidence:** 3

**Summary:**

This work investigates the effect of training with repeated examples (like training over the training data over multiple epochs) compared to training on very large datasets only once. The authors run experiments on three large-scale math problems, to compute GCD, modular multiplication, and computing eigenvalues using varying sizes of transformers and varying data and training budgets. The authors also propose a two-set training method in which they repeat one small subset of data throughout training and otherwise train on new data continually.

**Strengths:**

The paper is written clearly and is easy to follow and understand. The problem posed is important and if answered can be helpful in guiding training methods, fine-tuning and data gathering for complex problems. The results for the two-set training method bring up interesting questions about the role of memorization that would be interesting for future research.  All experiments are very thorough, showing good evidence for their claims on the datasets chosen.

**Weaknesses:**

From my point of view, this is not a particularly new idea. Training over many epochs over the training data is standard in most applications. This is perhaps not common in new applications such as in NLP as the authors cite for large language models. Given this, it would have been more interesting to see applications within NLP instead. The problems considered (algebraic problems) are very different, and the connection to other applications like NLP is not well-motivated.

In terms of related work, it may be interesting to investigate the connection to continual learning and catastrophic forgetting. This is more geared towards learning new tasks, and not about revisiting old examples from the same distribution. However, some of the work done here may provide valuable insight and would be nice to have some discussion comparing the methods in this area with yours.

Moreover, your idea of two-set learning sounds similar to spaced repetition in human learning -- being shown old instances right before you are about to forget them. There is not much work in this direction for ML training applications, although I did find this paper [1] which seems to have similar ideas to yours, proposing an order and frequency to the examples being trained on.


[1] Repeat before Forgetting: Spaced Repetition for Efficient and Effective Training of Neural Networks, Hadi Amiri, Timothy A. Miller, Guergana Savova.

**Questions:**

1. Why did you choose these experiments? I understand it is easier to control the data and the distribution of repeated examples. However, it is hard to see why conclusions drawn on algebraic problems should extend to other more natural data and NLP problems.

2. Can you try your ideas and run the experiments on non-synthetic datasets? It would be interesting to see if the two-set training works well for other data like CIFAR-10 or other natural datasets that you may find more appropriate.

3. Can you provide some discussion on continual learning etc. as I mentioned in the above section?

4. In Figure 5, the results are quite unstable as a function of your parameters, you need to be very careful in choosing the repeated set size etc. In practice, iterating over the possible set sizes and repeated set probability and training given each choice is very expensive. Do you have any intuition for how to choose these hyper-parameters in general? There does seem to be a general common "shape" where training works better.

---

> ### Author Response · Authors · 2024-11-19
> **Author's response (1/2)**
>
> We thank the reviewer for their time and effort and for acknowledging the importance and thoroughness of our work.
>
> >Weaknesses:
> From my point of view, this is not a particularly new idea. [...]
>
> Our observation are made on reasoning tasks, like mathematics, that are solved with *transformers*. Allow us to provide some more context here:
> Recent work on compute-optimal language models [Hoffman et al. 22] shows that many previously trained large language models could have attained better performance for a given compute budget by training a smaller model on more data. Most prior large language models have been trained for a single epoch [Komatsuzaki et al.'19 *"One epoch is all you need"*] and some work explicitly advocates against reusing data [Hernandez et al. 22]. [Muennighoff et al. NeurIPS 2023 Best Paper] undertake an extensive study of multi-epoch training for LLMs on natural data. They find that, while models trained for a single epoch consistently have the best validation loss per compute, differences tend to be insignificant among models trained for up to 4 epochs and do not lead to differences in downstream task performance (but surely not to any improvements). There is thus a deep-rooted belief in the transformer community that single-epoch training yields the best performance, and only when data is constrained some studies show that for a limited number of epochs (up to 4) can still yield some benefits (though not comparable to training on more data). Multi-epoch training is viewed as a poor proxy in attempts to attain the performance of single-epoch training if data was abundant. Repetition of training sets for transformers is viewed as a bug, not a feature, only to be employed when data is scarce.
>
> In controlled transformer experiments we challenge the *single-epoch paradigm*: not only is repeated training on smaller data budgets powerful competition to single epoch-training (for the same number of training steps); in several cases repeated training on a smaller set allows to *unlock* capacities that are unattainable with single epoch training on a much larger dataset. In some cases, increased repetition of a smaller set leads to emergent phenomena of learning.
>
> We can contemplate how our observations carry over to LLMs trained on natural *language* data (NLP), and how they translate to actionable insights. While they seem at odds with the current practice of seeing training data only once, they might indicate that under the hood duplication in the training corpora mimics our two-set approach. If this is the case, intentional scrutiny of the training corpus to identify how to deliberately  enact our observations could be beneficial for learning efficiency. And *fine-tuning corpora*,  are often curated and feature less repetition. We believe two-set training, and associated methods, may directly prove beneficial for fine-tuning LLMs. However, the focus of our work is to study these phenomena in a *controlled setting*, not directly in the wild. Please kindly see our General Response on this as well.
>
> >In terms of related work, it may be interesting to investigate the connection to continual learning and catastrophic forgetting. This is more geared towards learning new tasks, and not about revisiting old examples from the same distribution. However, some of the work done here may provide valuable insight and would be nice to have some discussion comparing the methods in this area with yours.
>
> There is a deep difference between continual learning and our approach. Continual learning focuses on *distribution shift*. Our setting, on the other hand, always remains *in distribution*. This said, our experiments in Section 6 and Appendix D.1 suggest that using selected examples in the repeated set, in a manner of curriculum learning, does not improve model performance. Please see also our General Response.
>
> >Moreover, your idea of two-set learning sounds similar to spaced repetition in human learning -- being shown old instances right before you are about to forget them. There is not much work in this direction for ML training applications, although I did find this paper [1] which seems to have similar ideas to yours, proposing an order and frequency to the examples being trained on.
>
> >[1] Repeat before Forgetting: Spaced Repetition for Efficient and Effective Training of Neural Networks, Hadi Amiri, Timothy A. Miller, Guergana Savova.
>
> As in the General Response, we would like to point out that the work [1], and a large body of interesting works on catastrophic forgetting deal with the scenario of *distribution* shift (or multiple consecutive tasks in the training set) - which is not the scenario of our work. Indeed, our observed phenomena are all the more baffling as they appear without any distribution shift.

---

> > ### Author Response · Authors · 2024-11-19
> > **Author's response (2/2)**
> >
> > Questions:
> > 1. Why did you choose these experiments? I understand it is easier to control the data and the distribution of repeated examples. However, it is hard to see why conclusions drawn on algebraic problems should extend to other more natural data and NLP problems.
> >
> > Please kindly see our General Response. As in the works described there, we view our results as a initial step to uncover a new phenomenon in a controlled setting. We will make this further clear in the paper.
> >
> >
> > 2. Can you try your ideas and run the experiments on non-synthetic datasets? It would be interesting to see if the two-set training works well for other data like CIFAR-10 or other natural datasets that you may find more appropriate.
> >
> > Our paper focuses on transformer architectures only, as commonly used in all contemporary NLP and most AI4Math applications. We would like to limit our scope outside of vision tasks especially when other architectures (like VGG or ResNet) come into play, to have a clearly delineated study. Indeed, an object of future study could be a vision transformer (to stay in the transformer landscape) - but that would likely be an overkill for CIFAR10 (which can be learned very well by simpler architectures). Please also see our General Response on precedence for controlled experiments on synthetic data.
> >
> > 3. Can you provide some discussion on continual learning etc. as I mentioned in the above section?
> >
> > Please see our General Response as it pertains to distribution shift in continual learning. Please note that we have a relatively lengthy related discussion on curriculum learning in Appendix C.1. and the extended list of experiments there. Unfortunately, due to length constraints, we could only fit a summary into the main body of the paper.
> >
> >
> >
> >
> > 4. In Figure 5, the results are quite unstable as a function of your parameters, you need to be very careful in choosing the repeated set size etc. In practice, iterating over the possible set sizes and repeated set probability and training given each choice is very expensive. Do you have any intuition for how to choose these hyper-parameters in general? There does seem to be a general common "shape" where training works better.
> >
> > Indeed, this depends on the task! For the GCD, two-sample training results in improved performance for a broad set of parameters. For modular addition, we empirically observe that the correct parameters lie on a hyperbola, which suggest a constant ratio between the repetition on the small and large sample.
> >
> >
> > We hope that we addressed all your concerns, and that you are satisfied with our responses and changes. Please consider raising your score.

---

### Official Review · Reviewer_knen · 2024-11-01

**Soundness:** 3
**Presentation:** 3
**Contribution:** 3
**Rating:** 6
**Confidence:** 3

**Summary:**

The present paper deals with the essential question of the role of repetition in the training data of machine learning models. More specifically, the work delves into the effect of introducing repeated examples when training transformers on three mathematical problems: finding the greatest common divisor between two numbers, performing modular multiplication, and computing the eigenvalues of (small) matrices. Through extensive numerical experiments, the authors show that repetition during training is very valuable for transformers to perform well on these tasks, and that might be necessary for learning to emerge in the case of modular multiplication. In light of these results, they propose a two-set training procedure, in which training examples may be drawn from either a large set of examples that will be seen only a handful of times or from a much smaller set of examples repeated many times during training. Consistent with their initial experiments, they find that this procedure improves the performance of trained networks for given data budgets, and in some cases enables learning. The authors finally consider variations of this procedure, notably the natural idea of curating the repeated set to further improve performance. Surprisingly, they find that curating does not provide further gains, or may be detrimental.

**Strengths:**

Overall, this paper explores an important aspect of machine learning—understanding why algorithms are usually expected to suffer from repetitions whereas human and animal learning appear to be highly dependent on them for learning—and could therefore be a significant contribution to this puzzle as it provides counter-examples that deserve to be better understood. By relying on the transformer architecture, which has become ubiquitous in many practical implementations, the authors also maximize the chance that their findings, and the training protocol they propose, may be taken advantage of in practical settings. The study of mathematical tasks provides a well-controlled environment where memorization can be identified, while properly learning an algorithm is necessary to achieve good generalization. The paper is well-written, and the author’s conclusions are clearly stated.

**Weaknesses:**

I believe the main limitation of this work lies in the limited number of tasks it considers, undermining the generality of its conclusions. In particular, the authors repeatedly make statements such as ‘models trained on smaller sets of repeated examples outperform models trained on larger sets of single-use examples’ (abstract); ‘smaller data budgets and more frequent repetition allow for faster learning, but also for much better performance’ (p. 5); ‘learning emerges through repetition’ (p. 5) etc. Based on the presented results however, it appears that these claims should be slightly tempered: there is a tradeoff between the repetition and the diversity that is necessary to avoid overfitting, as clearly illustrated in Fig. 2. While the authors show that repetitions at a given data budget may be beneficial, this relation is strongly non-monotonous, and in the greatest common divisor task one notably still requires a large number of unique samples to perform correctly (and as apparent in Fig. 4 the balance between repetition and diversity is hard to strike in this problem!). Clarifying this point and emphasizing that it is a tradeoff would make the paper more convincing, and would add to the relevance of the two-set procedure which precisely tries to provide the ideal tradeoff.

It also remains unclear whether the two-set procedure is relevant in settings where the transformer must not necessarily learn a clear-cut algorithm. For instance, could such a procedure be effective in image classification and more generally vision-related tasks? It seems plausible that the need for repetition is related to the hardness of the task at hand, which is not discussed in the paper.
In that respect, the difference in hardness of the three tasks exposed is not straightforward to understand: the authors notably state that the computation of eigenvalues is the hardest task they consider, yet the smallest model (in number of parameters) is used to tackle it. Therefore, it is not evident how the eigenvalue problem contributes to the clarity of the paper and its conclusion, also considering the relatively poor results that the authors find (4/30 as the maximum trained model success rate if I understand correctly) and the lack of clear plots dedicated to this third problem, despite it being expected to be the most difficult to tackle.

From a more technical standpoint, the accuracy metric chosen by the authors is not easy to interpret, notably in Fig. 2. While Appendix A provides some explanation as to how the chosen accuracy may stay fixed while the test loss explodes, the fact that it is on the eigenvalue computation and not the GCD problem for clear interpretation of Fig. 2 makes it still unsatisfactory. Besides, Fig. 6 does show some examples where the accuracy decreases as the model overfits.

Finally, the conclusion of the authors that the transformers should somehow be able to distinguish between already seen and unseen examples is puzzling. Could the role of repetition not be simply explained by the fact that there needs to be some form of symmetry breaking in the direction of the loss to follow and that repeating examples allows for a clear direction in the loss landscape to emerge? In that respect, I point the authors towards the papers:
Dandi, Y., Troiani, E., Arnaboldi, L., Pesce, L., Zdeborová, L., & Krzakala, F. (2024). The benefits of reusing batches for gradient descent in two-layer networks: Breaking the curse of information and leap exponents. arXiv preprint arXiv:2402.03220.
Arnaboldi, L., Dandi, Y., Krzakala, F., Pesce, L., & Stephan, L. (2024). Repetita iuvant: Data repetition allows sgd to learn high-dimensional multi-index functions. arXiv preprint arXiv:2405.15459

A more in-depth discussion of the different mechanisms that could explain the presented phenomenology, could significantly improve the paper.

**Questions:**

List of questions (including some repetitions of the above):
- Do the authors believe that their findings hold in tasks that do not require learning an algorithm? For example in vision-related tasks where memorization could be more useful (and sufficient in some cases)?
- Do they know of an example where repetition or the two-step procedure is not beneficial? One could notably think of tasks where curriculum learning is decisively effective, then shouldn’t random repetitions at least fare worse than curated ones? In other words, can the two-step procedure work well when curriculum learning works well?
- Can the two-step procedure be performed by showing only the repeated samples in a first learning phase, and then moving on to the more diverse samples in a second phase? i.e. focus first on discovering some ‘rules’, before generalizing these rules? I believe that finding good results in such a procedure would go in the direction of a loss-landscape-based interpretation of the phenomenon, in the sense of the Dandi et al. paper mentioned above. This would also further clarify the difference between what the authors observe and grokking (in addition to the different timescales and sample sizes they mention in Sec. 2).
- Can the authors clarify how accuracy is never affected by overfitting in the GCD task? How general do they expect this behavior to be?
- Can the authors clarify what they mean by a transformer being able to identify deja vu and jamais vu examples? Why should the architecture ‘care’?

**AFTER DISCUSSION**

The authors have addressed the main concerns I presented and included different explorations that I believe clarify and strengthen their message.  I will raise my score to a 6, for the moment.

---

> ### Author Response · Authors · 2024-11-19
> **Author's response (1/3)**
>
> We thank the reviewer for their extensive and thoughful comments, and in particular the strong and concise summary of the strengths of our paper.
>
> >I believe the main limitation of this work lies in the limited number of tasks it considers, undermining the generality of its conclusions. In particular, the authors repeatedly make statements such as ‘models trained on smaller sets of repeated examples outperform models trained on larger sets of single-use examples’ (abstract); ‘smaller data budgets and more frequent repetition allow for faster learning, but also for much better performance’ (p. 5); ‘learning emerges through repetition’ (p. 5) etc. Based on the presented results however, it appears that these claims should be slightly tempered: there is a tradeoff between the repetition and the diversity that is necessary to avoid overfitting, as clearly illustrated in Fig. 2. While the authors show that repetitions at a given data budget may be beneficial, this relation is strongly non-monotonous, and in the greatest common divisor task one notably still requires a large number of unique samples to perform correctly (and as apparent in Fig. 4 the balance between repetition and diversity is hard to strike in this problem!). Clarifying this point and emphasizing that it is a tradeoff would make the paper more convincing, and would add to the relevance of the two-set procedure which precisely tries to provide the ideal tradeoff.
>
> Please see our comments in the General Response that address the generality of our conclusion. There we provide arguments and ample precedence on doing experiments in a controlled (and thus necessarily limited) setting.
>
> As for the formulation of the claims in our paper, we were aiming to present the facts and not to overclaim. As such, we had hoped that our writing was measured and truthful:
> * In the abstract, the sentence you quote begins with "on three problems of mathematics (...) we show..."
> * On page 5, these sentence appear while summarizing the result of a specific task, we make no claim about their generality.
> * In the discussion, we devote a paragraph to explain that extending our findings to LLMs is the subject of further study
>
> Please let us know which claims you are uncomfortable with and we will do our best to present them in a measured way with appropriate qualifications.
>
> >It also remains unclear whether the two-set procedure is relevant in settings where the transformer must not necessarily learn a clear-cut algorithm. For instance, could such a procedure be effective in image classification and more generally vision-related tasks? It seems plausible that the need for repetition is related to the hardness of the task at hand, which is not discussed in the paper. In that respect, the difference in hardness of the three tasks exposed is not straightforward to understand: the authors notably state that the computation of eigenvalues is the hardest task they consider, yet the smallest model (in number of parameters) is used to tackle it. Therefore, it is not evident how the eigenvalue problem contributes to the clarity of the paper and its conclusion, also considering the relatively poor results that the authors find (4/30 as the maximum trained model success rate if I understand correctly) and the lack of clear plots dedicated to this third problem, despite it being expected to be the most difficult to tackle.
>
> This points to a curious fact about math transformers: the tasks they struggle to learn are not necessarily the hardest from a mathematical point of view. For instance, Charton (Linear Algebra with Transformers, in our references), shows that even small transformers have no difficulty learning to compute eigenvalues or vectors for real symmetric matrices of dimension up to 8x8. On the other hand, modular multiplication, a much easier mathematical task, cannot be learned without repeated samples, therefore proving hard for transformers.
>
> We agree that the eigenvalue experiments are less spectacular: repetition only brings a small improvement in performance. We believe this is because this task is, in fact, *easier* for transformers to learn.
>
> Still, we decided to add it to the paper, because it is very different from the two others:
> * it features real/floating point numbers instead of integers
> * it is a more complicated mathematical task, featuring an approximate algorithm
> * it uses noisy data (due to the rounding of real numbers into floats)
>
> We chose to inlcude these experiments, because they suggest our conclusions extend beyond arithmetic on integers. We will clarify this in the final version.

---

> > ### Author Response · Authors · 2024-11-19
> > **Author's response (2/3)**
> >
> > >Finally, the conclusion of the authors that the transformers should somehow be able to distinguish between already seen and unseen examples is puzzling. Could the role of repetition not be simply explained  [...]
> > >A more in-depth discussion of the different mechanisms that could explain the presented phenomenology, could significantly improve the paper.
> >
> > We agree with you that the next step is to understand the mechanism behind the phenomenon we have exhibited. Indeed we have tried to come up with a toy model that would help us explore this. However, we believe our most crucial ablation in the case of the two-set effect is when we study mono-batches - taking batches either entirely from the "frequent" set or the "infrequent" one. It was curious to us to discover that the effect disappears when mono-batches are used, even if we maintain the same sample-frequency for both sets. This - perhaps somewhat disappointingly - kills several attempts to explain the effect, since it seems to depend on the optimization procedure. It also precludes explanations from a "curriculum learning" point of view.
> >
> > However, we are very grateful to you for pointing out the related very recent work of [Dandi et al. 24] and [Arnaboldi et al. 24] - we will add them to our paper (though please note that especially the more relevant latter one appeared relatively close to the submission deadline of this paper and by the rules of ICLR does not necessarily need to be considered as they appeared while our work was ongoing). Note, however, that our mono-batch ablation immediately precludes the explanation presented in [Arnaboldi et al. '24]. There, it is argued that batches need to be seen more than once (in the restricted case of the the multi-index model - which is a significant restriction - for two-layer nets) for beneficial symmetry breaking to happen. Our mono-batch ablation shows that this cannot explain what we observe. First, in our case, no two batches are the same, as data gets reshuffled. However, mono-batches are more likely to resemble each other when they come from the smaller, frequent set, compared to the multi-batch case where by design most examples in each batch will never be seen again (in the case of unlimited data at least). This means that our mono-batch case should resemble the scenario in [Arnaboldi et al. 24] more - yet it fails to show the observed effect. We conclude that the mechanism of [Arnaboldi et al. 24] cannot be at the origin of our observed effect.
> >
> > We can include this discussion in the appendix, and we could try to find other mechanisms that do *not* explain what we see, but we see limited interest in discussing what doesn't work (and there isn't a lot of works on this kind of repetition anyway). However, we are happy to include what you might consider relevant - please let us know your suggestions if so.
> >
> > >Questions:
> > - Do the authors believe that their findings hold in tasks that do not require learning an algorithm? For example in vision-related tasks where memorization could be more useful (and sufficient in some cases)?
> >
> > For the three settings we have chosen, it is not clear to us that the model is learning an *algorithm*, here. For the GCD, transformers have been shown to learn to classify their entries, by learning divisibility rules, which are very different from the Euclid algorithm. We believe our findings apply to reasoning tasks, with deterministic answers. We have not extended our analysis to vision tasks, especially when other architectures (like VGG or ResNet) come into play, as we wanted to remain in the controlled settings of transformers.
> >
> > - Do they know of an example where repetition or the two-step procedure is not beneficial? One could notably think of tasks where curriculum learning is decisively effective, then shouldn’t random repetitions at least fare worse than curated ones? In other words, can the two-step procedure work well when curriculum learning works well?
> >
> > Curriculum learning, learning from a particular distribution (log-uniform operands and outcomes) was shown to be beneficial for the GCD task (Charton 2024). In the beginning of Section 6 and Appendix D.1, we show that curating the repeated sample does not improve two-sample training performance. To be specific, we show:
> > 1. Prior work [Charton 2024] on solving GCD with transformers shows that a form of curriculum learning/data curation, namely training on log uniform operands, does improve performance
> > 2. The best performance of training with log-uniform operands is about the same as our proposed two sample training
> > 3. One cannot improve performance by mixing the two, i.e. by curating the small, more frequently seen subset, to have log uniform operands
> >
> > We have no comparison baseline for modular multiplication nor eigenvalue computation, which to our knowledge were not studied in prior work on curriculum learning.
> >
> > Please let us know if this answers your question.

---

> > > ### Author Response · Authors · 2024-11-19
> > > **Author's response (3/3)**
> > >
> > > - Can the two-step procedure be performed by showing only the repeated samples in a first learning phase, and then moving on to the more diverse samples in a second phase? i.e. focus first on discovering some ‘rules’, before generalizing these rules? I believe that finding good results in such a procedure would go in the direction of a loss-landscape-based interpretation of the phenomenon, in the sense of the Dandi et al. paper mentioned above. This would also further clarify the difference between what the authors observe and grokking (in addition to the different timescales and sample sizes they mention in Sec. 2).
> > >
> > > Thank you for this suggestion. Following your request, we have run this experiment, for both the GCD and modular multiplication task. We first train the model on a repeated set (or 50k and 2.5M examples respectively), for a fixed number of epochs, then switch to a larger set (of 100M examples). For each task, we ran 50 experiments: 10 model initializations, and 5 levels of repetition. All models overfit very rapidly, causing lower performance than previously observed (8 correct GCD predicted, vs 13 in previous experiments). Once the model switches to the large training set, catastrophic forgetting sets in, and the model seems to learn from scratch. We note that this second phase is slightly delayed: overfitting seems to result in a bad initialization of weights.
> > >
> > > We believe these results are accounted for by our observation on mini-batches (Section 6 and Appendix D.2). If repeated and non repeated examples are presented separately, i.e. not mixed into the same mini-batches, the benefits of two-sample training disappear. However, we are happy to include the above experiments in the paper if you feel they add value.
> > >
> > > - Can the authors clarify how accuracy is never affected by overfitting in the GCD task? How general do they expect this behavior to be?
> > >
> > > Thank you for this question. In the GCD tasks, we did observe a small drop in accuracy due to overfitting for models trained on very small datasets (less than 500k examples). We believe this is due to the fact that models learn strong and deterministic rules (see Charton 2024), which are very difficult to "unsee". We believe this is characteristic of deterministic tasks.
> > >
> > >
> > > - Can the authors clarify what they mean by a transformer being able to identify deja vu and jamais vu examples? Why should the architecture ‘care’?
> > >
> > > The point we tried to make is the following: The model does "care" about deja vu (seen before) and jamais vu (never seen), or the results of Sections 4 and 5 would not exist. What we learn in Section 4 is that there is a qualitative difference between model trained on repeated examples (even with low repetition) and models trained on unique (or almost unique) examples. For such a difference to exist, the model has to be able to "tell", somehow, that examples were repeated. We do not have a substantiated running hypothesis on how this happens, and believe it is a thought proviking subject for future research.
> > >
> > >
> > > We hope our additional explanations and the experiments we have now performed upon your suggestion clarify things. We would be grateful if you could let us know what you find lacking and raise your score if this finds your approval.

---

> ### Comment · Reviewer_knen · 2024-11-24
> **Response**
>
> (1/3)
>
> **which claims you are uncomfortable with** I realize that my phrasing was perhaps a bit unclear since I combined two different remarks in a single comment. I agree with the authors on their general response regarding the ‘issue’ of generality. Rather, and this was the second point of my original comment, I did get the feeling that their conclusions were, where I quoted them, perhaps a bit too schematic and simplified, and that they might not sufficiently emphasize the fact that there is always a tradeoff between diversity and repetitions. To be specific, I did not get the feeling that the phrase ‘smaller data budgets and more frequent repetition allow for faster learning’ was sufficient to summarize the situation that they present: this is only true when there is already a very significant number of unique examples and overfitting is an issue that they discuss. As mentioned in my original comment, I believe that Fig. 4 for example is particularly interesting in that aspect, as the balance between repetition and diversity is evidently hard to find on some problems. Note that I find this subtle balance most interesting, which is also why I believe it is detrimental to not highlight it more clearly in the abstract, introduction and summaries.
>
> **inlcude these experiments...they suggest our conclusions extend beyond arithmetic on integers** I thank the authors for the clarification, and encourage them to highlight this curious fact about math transformers in addition to stating their motivation for the eigenvalue experiments more clearly.
>
> (2/3)
>
> **However, we are happy to include what you might consider relevant** I thank the reviewers for their complete answer. I believe that mentioning several possible explanations, including this one, and how likely they believe them to be could only improve their work. In my opinion including the complete discussion on this point in the appendix would be a welcome addition.
>
> **We have not extended our analysis to vision tasks...** I definitely think that including something like ‘We believe our findings apply to reasoning tasks, with deterministic answers’ explicitly in the paper would be beneficial. Also mentioning the fact that transformers learn the GCD task in such a way could be helpful to provide further context for readers unfamiliar with these type of mathematical tasks in transformers.
>
> **Please let us know if this answers your question.** I thank the authors for their answer, which I believe clearly addresses the question. Do they have an understanding of why these two approaches cannot be combined and how they may coexist? Is there any chance that the transformer implementation ends up being significantly different between the two training strategies, which could explain why they are mutually exclusive? Overall I encourage them to suggest some scenarios if they have any, even without evidence, as I think these observations are most interesting but have a hard time understanding how they may emerge. I do appreciate the authors explicitly stating the counter-intuitive nature of this observation.
>
> (3/3)
>
> **we are happy to include the above experiments in the paper** I thank the authors for running this experiment given the limited timeframe of the rebuttal period. I think it could be interesting to include it in the appendix where the Dandi et al. scenario is discussed, at least stating that this was attempted.
>
> **We believe this is characteristic of deterministic tasks.** I thank the author for their reply. Again I think that mentioning this interpretation (even e.g. in a footnote) could be valuable, as I had not seen this behavior as clearly before.
>
> **We do not have a substantiated running hypothesis on how this happens...** I am still confused on this point. To me, the results of Sections 4 and 5 do not necessarily require the transformer to act differently at the level of individual inputs, which would be the case if the transformer ‘cared’ about deja vu vs jamais vu if I understand correctly. To make this statement, I feel like further investigation on input-specific properties of the model would be required. I understand that ‘we learn in Section 4 is that there is a qualitative difference between model trained on repeated examples (even with low repetition) and models trained on unique (or almost unique) examples’, but I still don’t see why this could not be due only to some difference emerging in the learning dynamics and not in the transformer representation per se.
>
>
> All in all, I thank the authors for their very complete response and for performing the experiment I had suggested — I also apologize for this quite late response on my side. I look forward to their revised version implementing the feedback from all reviewers and I believe that I should be able to raise my score appropriately.

---

> > ### Author Response · Authors · 2024-11-25
> > **Thank you for the response, new version uploaded with your feedback**
> >
> > We would like to thank this reviewer for their engagement, for their thoughtfulness and for their effort in helping us improve our paper. In response, we have modified our manuscript with all changes marked in blue and outlined in a new General response. To your specific points:
> >
> > >**which claims you are uncomfortable with** ...  To be specific, I did not get the feeling that the phrase ‘smaller data budgets and more frequent repetition allow for faster learning’ was sufficient to summarize the situation that they present: this is only true when there is already a very significant number of unique examples and overfitting is an issue that they discuss. ...
> >
> > We have tried to give this point justice without overflowing the 10 pages and cutting the flow. We hope the addition to the caption of our main illustrative figure expresses this better now.
> >
> >
> > >**inlcude these experiments...** they suggest our conclusions extend beyond arithmetic on integers
> >
> > Following your suggestion, we have included additional language to this effect in the part about eigenvalues in Section 3.
> >
> >
> > (2/3)
> >
> > >**However, we are happy to include what you might consider relevant**  I believe that mentioning several possible explanations, including this one, and how likely they believe them to be could only improve their work. In my opinion including the complete discussion on this point in the appendix would be a welcome addition.
> >
> > Following your suggestion, we have added an entire Appendix A to the paper which includes this discussion and other discussion topics we have touched upon in the rebuttal discussion. Thank you for helping us improve our paper.
> >
> >
> > >**We have not extended our analysis to vision tasks... I definitely think that including something like ‘We believe our findings apply to reasoning tasks, with deterministic answers’** explicitly
> >
> >
> > We have changed our discussion section in the main body of the paper to add this point very clearly. We have also moved the more ambigious piece on deja vu and jamais vu to the discussion Appendix A
> >
> >
> > >**Please let us know if this answers your question.** I thank the authors for their answer, which I believe clearly addresses the question. Do they have an understanding of why these two approaches cannot be combined and how they may coexist? Is there any chance that the transformer implementation ends up being significantly different between the two training strategies, which could explain why they are mutually exclusive? Overall I encourage them to suggest some scenarios if they have any, even without evidence, as I think these observations are most interesting but have a hard time understanding how they may emerge. I do appreciate the authors explicitly stating the counter-intuitive nature of this observation.
> >
> > We have added clarifying notes to Appendix D.1 to make the summary on GCD learning more clear.
> >
> > And indeed, it might be possible to combine these two training strategies (data distribution shift in the training data and two-set training). We have launched an additional experiment that might show that such a combination could be possible. However, we feel this would be the subject of an entirely new study and would like to confine this paper to the effect we observe there, without combination, for a first paper on this topic. We thanks this reviewer for their very thought provoking suggestion.
> >
> > (3/3)
> >
> > >**we are happy to include the above experiments in the paper** I thank the authors for running this experiment given the limited timeframe of the rebuttal period. I think it could be interesting to include it in the appendix where the Dandi et al. scenario is discussed, at least stating that this was attempted.
> >
> > We have added an entirely new Appendix D.6 with the other ablation experiment to describe this experiment. We refer to in in the new discussion Appendix A.
> >
> >
> > >**We believe this is characteristic of deterministic tasks**... Again I think that mentioning this interpretation (even e.g. in a footnote) could be valuable, as I had not seen this behavior as clearly before.
> >
> > We hope you feel this is now adressed in our main discussion (page 10).
> >
> >
> > >**We do not have a substantiated running hypothesis on how this happens**... ...
> >
> > We have removed this somewhat speculative discussion on deja vu and jamais vu from the main body discussion in the paper, as indeed it might cause misinterpretation.
> >
> > >All in all, I thank the authors for their very complete response and for performing the experiment I had suggested — .... I look forward to their revised version implementing the feedback from all reviewers and **I believe that I should be able to raise my score appropriately.**
> >
> > We are extremely grateful for the thorough and thoughtful feedback and for the appreciation of our efforts to improve the paper in response to the feedback we received. We truly hope the changes in the paper and our responses have removed some of your hesitations and hope you will now be able to raise your score. Thank you!

---

### Official Review · Reviewer_yjc8 · 2024-11-02

**Soundness:** 3
**Presentation:** 3
**Contribution:** 3
**Rating:** 6
**Confidence:** 3

**Summary:**

This paper empirically shows that the repetition of training data can be beneficial in certain setups. The authors conduct experiments on three algorithmically generated datasets, and show that model trained with small data budgets outperforms model trained on larger data budgets. The authors also identify a two-phase training algorithm that gives faster training and better performance.

**Strengths:**

1. The experiments are conducted on three algorithmically generated datasets of math tasks, which is an ideal setup of controlled experiments. The experiment setups are clearly described, and the results are well presented and explained.

2. The empirical findings of the beneficial of repeated examples have decent practical meanings. The success of two set training is interesting and novel to me.

**Weaknesses:**

1. My major concern is that the experiments are only conducted on algorithmically generated synthetic data, instead of real world datasets. This makes me not fully convinced that the experimental findings are universal and can be transferred to more realistic setups.
Specifically, I am wondering whether the main findings, especially the success of two set training, also applies to real language datasets. Do you have empirical results to support that? Besides, the three dataset are all math problems. Do you think that this specific type of training data might have a positive influence on the performance of repeated data?

2. Does the emergent property in the title mean that the observed phenomena only happens for training at scale? Do you have found any criteria that can indicate under what circumstance can repeated data or two-set training be beneficial?

**Questions:**

See weaknesses part.

---

> ### Author Response · Authors · 2024-11-19
> **Author's response**
>
> We thank the reviewer for taking time to read our paper, for their valuable comments and the overall positive reception of our work. We would like to address your concerns below.
>
> Weaknesses:
> 1. My major concern is that the experiments are only conducted on algorithmically generated synthetic data, instead of real world datasets. This makes me not fully convinced that the experimental findings are universal and can be transferred to more realistic setups. Specifically, I am wondering whether the main findings, especially the success of two set training, also applies to real language datasets. Do you have empirical results to support that? Besides, the three dataset are all math problems. Do you think that this specific type of training data might have a positive influence on the performance of repeated data?
>
>
>
> Please see our General Response that addresses your concern - thank you!
>
> 2. Does the emergent property in the title mean that the observed phenomena only happens for training at scale? Do you have found any criteria that can indicate under what circumstance can repeated data or two-set training be beneficial?
>
> Thank you for pointing out that "emergence" is an overloaded term. In this paper, we define emergence as the fact that a property is only learned in the particular case, i.e. with repeated examples. We do not discuss the impact of scaling (on the GCD, Charton 2024 mentions that scaling has a very limited impact on model performance). But we agree that there is another conotation of "emergence" in the community that refers to phenomena appearing only *at scale*. We will disambiguate this in the next version of our paper; thanks for pointing this out.
>
> We hope we have adressed your questions to your satisfaction.

---

### Author Response · Authors · 2024-11-19
**General Response: Synthetic Data Setting, Relation to Continual/Curriculum Learning, Additional Experiments**

1/2
We thank the reviewers for taking time to read our paper and for their valuable comments and are happy to see the uniform interest our observed phenomena have elicited. We have responded to each reviewer individually and can include some of the desired discussion/references in the paper - we plan to upload a suggested new version after gathering initial feedback and will notify when we do.

**Synthetic Data Setting:**
Several reviewers have commented on the nature of our experiments on synthetic data only. We would like to address this point here:

We have made a deliberate choice to use synthetic data in this work: on "real world data" (i.e. text scraped off the Internet, or code from public repositories), repetition cannot be controlled, and it would not be possible to measure the effects observed here in a meaningful way. We believe two-set training is somehow "baked into" many natural language tasks, and may even explain some of the success of language models, because the training data is replete with repeated sentences and paragraphs (or statements and functions, in the case of code). The mathematical nature of the task is only visible to human observers: for the model, which minimizes a cross-entropy loss, it is not different from other "letter games", such as code refactoring, text summarizing, or various reasoning tasks. As such, we do not think it has an impact on our conclusion.

We would like to add that the practice of studying phenomena in Deep Learning, and even discovering new ones, on *synthetic* data is widespread and established. Allow us to provide just three examples:
- the influential "grokking" paper [Power et al.22] uses small, synthetic, mathematical datasets (among them modular multiplication) to exhibit what is now well known as "grokking"
- in-context learning of transformers has gained its theoretical foundation in an early, highly cited paper [Garg et al. 22] (with many follow-ups). Here, the synthetic data comes from linear functions, but the insights on how "meta" learning works in-context are profound.
- another example is Zeyuan Allen-Zhu's impressive series of works (with various co-authors) on the "physics of deep learning" (cf. the excellent and very well attended NeurIPS'24 tutorial he gave on this subject). All the experiments performed there are on controlled, synthetic data, with interesting effects emerging - as in our case.

All of these papers hope and speculate that the insights they gain in these controlled settings carry over to explain "real" phenomena observed in larger models, at least to a certain extend, and it seems they have proven right, even though they have not directly performed "real world" studies. Note that there is increased visibility for this approach, as witnessed by the upcoming workshop at NeurIPS'24 on Scientific Methods for Understanding Deep Learning (ScifoDL), which precisely *promotes a complementary approach that is centered on the use of the scientific method, which forms hypotheses and designs controlled experiments to test them.*

Complementary to fostering understanding or discovery of new phenomena that carry over to larger realistic models, there is an additional important aspect of our work that pertains to AI4Math. There is increasing literature on using transformers for Math problems and success stories start to emerge (see e.g. [Alfarano et al. 24, Charton et al.24] and [New Scientist 24]), based on transformer models of the kind we utilize in our study. For this community, our insights are of immediate relevance, as they give prescriptive advice on how to utilize training data (iterate rather than one-pass, use two-set training). Indeed, one feature of our tasks - which is the case in most AI4Math settings - is that they are deterministic : there is only one correct solution. One might speculate how much this fact might impact our conclusions, which we believe is an excellent topic of study in future work. But even in this setting, we hope that our results will be of practical and immediate relevance.

[Power et al. 22] Alethea Power, Yuri Burda, Harri Edwards, Igor Babuschkin, Vedant Misra, "Grokking: Generalization Beyond Overfitting on Small Algorithmic Datasets", OpenAI, arXiv 2022

[Garg et al. 22] Shivam Garg, Dimitris Tsipras, Percy Liang, Gregory Valiant, "What Can Transformers Learn In-Context? A Case Study of Simple Function Classes", NeurIPS 2022

[Alfarano et al. 24] A Alfarano, F Charton, A Hayat, "Global Lyapunov functions: a long-standing open problem in mathematics, with symbolic transformers", arXiv 2410.08304

[New Scientist 24] New Scientist "Meta AI tackles maths problems that stumped humans for over a century" https://www.newscientist.com/article/2452780-meta-ai-tackles-maths-problems-that-stumped-humans-for-over-a-century/

[Charton et al. 24] F Charton, JS Ellenberg, AZ Wagner, G Williamson, "PatternBoost: Constructions in Mathematics with a Little Help from AI", arXiv 2411.00566

---

> ### Author Response · Authors · 2024-11-19
> **General Response (cont)**
>
> 2/2
>
> **Relation to Continual Learning/Catastrophic Forgetting:**
> Two reviewers have also asked about the connection to *continual learning*. We would like to point out that continual learning refers to a setting with *distribution shift* : the learner is supposed to adjust to new distributions in the input - and to avoid catastrophic forgetting of earlier tasks. This is an extremely interesting research area - but it is not what we study. In our setting the *distribution is always the same*, both for repeated examples (Section 4) and for the new phenomenon of two-set training (Section 5). Both the more frequently sampled and the less frequent larger set come from exactly the same underlying distribution - which is what makes this so baffling to us. We hope this clarifies why we have not drawn more parallels to continual learning/catastrophic forgetting.
>
> **Curriculum Learning:** Several reviewers brought up curriculum learning, where training data is presented in a particular order (usually from easy to hard). Our work is different from curriculum learning:
> We show that *randomly* selecting a small subset of the training data, and repeating them more often can significantly enhance performance or even overcome learning bottle necks. We discover a synergistic effect: neither training on the small set alone, nor training with unlimited data budget in one epoch would allow any learning at all - it is the combination of both that makes two-set training powerful! The fact that the repeated set can be chosen at random, and that curating repeated examples brings no improvement in performance sets it aside from curriculum learning and suggest that what matters, here, is seeing the *exact same* example several times.
>
> **Additional Experiments** In response to a request by reviewer knen, we have performed additional experiments for a different curriculum learning variant in connection with two-set sampling: we combine (some form of) curriculum learning with two-set training to show that training on the small, frequent set first before switching to the larger set does not provide any benefits (at best) and slows down learning (at worst).

---

### Author Response · Authors · 2024-11-25
**Updated manuscript**

In response to the reviewers' feedback and in particular the very helpful suggestions of reviewer knen, we have uploaded a modified version of our manuscript with changes highlighted in blue.

Summary of changes and additions:

1. We have clarified and expanded the eigenvalue experiments and a new Table 3 (p. 8-9) and moved them into the main body of the paper (eliminated an Appendix)
2. We have added a new Appendix A with additional discussion and references. In particular, it adds additional clarifications and comparison to curriculum learning and continual learning, as well as comparison to a couple theoretical works and discussion on how they do or do not apply to our observed effects.
3. We have added a little more discussion in the eigenvalue paragraph in Section 3 discussing the choice of this problem.
4. We have changed the final discussion (page 10) by being specific that our effects are observed for three deterministic reasoning tasks. We have also moved the more speculative 'deja-vu' paragraph to the Appendix A to make the discussion more crisp.
5. Some clarifications in Appendix D.1 and minor changes (marked in blue)

We hope the reviewers are satisfied with these changes and thank them for their constructive feedback.

---

### Note · Authors · 2025-01-28

I have read and agree with the venue's withdrawal policy on behalf of myself and my co-authors.